

# Reconstruction of the Greenland Ice Sheet surface mass balance and the spatiotemporal distribution of freshwater runoff from Greenland to surrounding seas

SEBASTIAN H. MERNILD

*Nansen Environmental and Remote Sensing Center, Bergen, NORWAY, Direction of Antarctic and Sub-Antarctic Programs, Universidad de Magallanes, Punta Arenas, CHILE, and Faculty of Engineering and Science, Western Norway University of Applied Sciences, Sogndal, NORWAY*

GLEN E. LISTON

*Cooperative Institute for Research in the Atmosphere, Colorado State University, Fort Collins, Colorado, USA*

ANDREW P. BECKERMAN

*Department of Animal and Plant Sciences, University of Sheffield, UK*

JACOB C. YDE

*Faculty of Engineering and Science, Western Norway University of Applied Sciences, Sogndal, NORWAY*

**Corresponding author address:**

Sebastian H. Mernild, e-mail: sebastian.mernild@nersc.no





**Abstract**
Knowledge about variations in runoff from Greenland to adjacent fjords and seas is important for
the hydrochemistry and ocean research communities to understand the link between terrestrial
and marine Arctic environments. Here, we simulate the Greenland Ice Sheet (GrIS) surface mass
balance (SMB), including refreezing and retention, and runoff together with catchment-scale
runoff from the entire Greenland landmass ($n$ = 3,272 simulated catchments) throughout the 35-
year period 1979–2014. SnowModel/HydroFlow was applied at 3-h intervals to resolve the
diurnal cycle and at 5-km horizontal grid increments using ERA-Interim (ERA-I) reanalysis
atmospheric forcing. Simulated SMB was low compared to earlier studies, whereas the GrIS
surface conditions and precipitation were similar. Variations in meteorological and surface ice
and snow cover conditions influenced the seasonal variability in simulated catchment runoff;
variations in the GrIS internal drainage system were assumed negligible and a time-invariant
digital elevation model was applied. Approximately 80 % of all catchments showed increasing
runoff trends over the 35 years, with on average relatively high and low catchment-scale runoff
from the SW and N parts of Greenland, respectively. Outputs from an Empirical Orthogonal
Function (EOF) analysis were combined with cross-correlations indicating a direct link (zero lag
time) between modeled catchment-scale runoff and variations in the large-scale atmospheric
circulation indices North Atlantic Oscillation (NAO) and Atlantic Multidecadal Oscillation
(AMO). This suggests that natural variabilities in AMO and NAO constitute major controls on
catchment-scale runoff variations in Greenland.
**KEYWORDS:** Empirical Orthogonal Function; Greenland freshwater runoff; Greenland Ice
Sheet; HydroFlow; Modeling; NASA MERRA; SnowModel; surface mass-balance



## 1. Introduction

The Greenland Ice Sheet (GrIS) is highly sensitive to changes in climate (e.g., Box et al. 2012; Hanna et al. 2013; Langen et al. 2015; Wilton et al. 2016; AMAP 2017). It is of scientific interest and importance because it constitutes a massive reserve of freshwater that discharges to adjacent fjords and seas (Cullather et al. 2016). Runoff from Greenland influences the sea surface temperature, salinity, stratification, marine ecology, and sea-level in a number of direct and indirect ways (e.g., Rahmstorf et al. 2005; Straneo et al. 2011; Shepherd et. al. 2012; Weijer et al. 2012; Church et al. 2013; Lenaerts et al. 2015).

The GrIS surface mass balance (SMB) and freshwater runoff have changed over the last decades and most significantly since the mid-1990s (e.g., Church et al. 2013; Wilton et al. 2016). For example, recent estimates by Wilton et al. (2016) showed a decrease in SMB from ~350 Gt yr$^{-1}$ (early-1990s) to ~100 Gt yr$^{-1}$ (late-2000s) and an increase in runoff from ~200 Gt yr$^{-1}$ (early-1990s) to ~450 Gt yr$^{-1}$ (late-2000s). For 2009 through 2012, the runoff has been estimated to include approximately two-third of the gross GrIS mass loss (Enderlin et al. 2014), while the net GrIS mass loss, on average, was 375 Gt yr$^{-1}$ (2011–2014) (AMAP 2017). The contribution of GrIS mass loss to global mean sea-level was around 5 % in 1993, and more than 25 % in 2014 (Chen et al. 2017). Noël et al. (2017), however, estimated the GrIS and peripheral glaciers to contribute approximately 43 % to the contemporary sea-level rise.

Runoff from the GrIS is an integrated response of rain, snowmelt, and glacier melt and other hydrometeorological processes (e.g., Bliss et al. 2014). Tedesco et al. (2016) estimated a 1979–2016 change in GrIS spatial surface melt extent of ~15,820 km$^2$ yr$^{-1}$, and a change in surface ablation duration of ~30–40 days in NE and 15–20 days along the west coast. At higher GrIS elevations, surface melt does not necessarily equal surface runoff because meltwater may





refreeze in the porous near-surface snow and firn layers (Machguth et al. 2016) where the firn
pore space provides potential storage for meltwater (Haper et al. 2012; van Angelen et al. 2013).
Melt water percolation, refreezing, and densification processes are common in GrIS snow, firn,
and multi-year firn layers – especially where semipermeable or impermeable ice layers are
present (Brown et al. 2012; van As et al. 2016). Such physical mechanisms and conditions in the
firn and multi-year firn layers lead, e.g., to non-linearity in meltwater retention (Brown et al. 2012).

The GrIS internal drainage system has received increased attention in recent years. This

is, in part, because the summer acceleration of ice flow is controlled by supraglacial meltwater
draining to the subglacial environment (Zwally et al. 2002; van de Wal et al. 2008; Shephard et
al. 2009). Enhanced production of supraglacial meltwater results in more water supplied to the
glacier bed, leading to reduced basal drag and accelerated basal ice motion. This process is
referred to as basal lubrication, and it constitutes a potential positive feedback mechanism
between climate change and sea-level rise (Hewitt 2013). At high GrIS elevations, surface
meltwater primarily drains to the glacier bed via hydrofractures (van der Veen 2007), whereas
meltwater is routed to the glacier bed via crevasses and moulins in the peripheral areas (Banwell
et al. 2016; Everett et al. 2016; Koziol et al. 2017). Rapid drainage of large volumes of GrIS
meltwater come from sudden release from supraglacial and proglacial lakes (known as a glacial
lake outburst flood (GLOF) or jökulhlaup), which are particular common in West Greenland
(Selmes et al. 2011; Carrivick and Quincey 2014). The seasonal evolution of the structure and
efficiency of the drainage system beneath the GrIS is indirectly assumed from our understanding
of the subglacial hydraulic potential beneath Alpine glaciers. This general understanding is used
explain the observed seasonal changes in ice motion (Bartholomew et al. 2010, 2012) where few
direct observations exist (Kohler et al. 2017). In fact, we know very little about spatiotemporal



shifts in the configuration of the subglacial drainage network beneath the GrIS. We therefore
assume that the subglacial drainage network in the natural system is dynamic and sensitive to
rerouting of water flow between adjacent catchments (so-called water piracy; Chu et al. 2016),
although we do not understand the details sufficiently to implement them in a runoff routing
model.
We also lack high resolution information on the spatiotemporal distribution of GrIS and
Greenland freshwater runoff to the fjords and seas, and the spatiotemporal distribution of solid-
ice discharge (calving) from tidewater glaciers is also largely unknown (Howat et al. 2013). To
address this lack of knowledge, information about the quantitative discharge (runoff and solid-
ice discharge) conditions from the numerous of catchments in Greenland is required. Available
GrIS calving rates are insufficient to represent the calving rates from the entire Greenland and
are therefore not generally included in overall Greenland freshwater estimates (Nick et al. 2009;
Lenaerts et al. 2015). This is an unaddressed gap, which likely prevents us from
comprehensively understanding the terrestrial freshwater discharge to the fjords and seas. This
also limits the subsequent the link between changes in terrestrial inputs and changes in the
hydrographic and circulation conditions. This unaddressed knowledge gap has further
implications for ocean model simulations, where, for example, earlier representations of
Greenland discharge boundary conditions were either non-existent or overly simplistic (e.g.,
Weijer et al. 2012).
Previous GrIS studies constructed a section-wise runoff distribution by dividing the ice
sheet into six to eight overall defined sections (e.g., Rignot et al. 2008; Bamber et al. 2012;
Rignot and Mouginot 2012; Lenaerts et al. 2015; Wilton et al. 2016). These studies illustrated an





increase in runoff since 1870 for all GrIS sections, with the greatest increase in runoff since mid-
1990s and in the SW part of the ice sheet.

Mernild and Liston (2012) reconstructed the GrIS SMB and the Greenland

spatiotemporal runoff distribution from ~3,150 individually simulated catchments, at 5-km
spatial, and daily temporal, resolutions covering the period from 1960 through 2010. Automatic
weather stations located both on and off the GrIS were used for atmospheric forcings, and the
study was carried out using a full energy balance, multi-layer snowpack and snow distribution,
and freshwater runoff model and software package called SnowModel/HydroFlow (Liston and
Elder 2006a; Liston and Mernild 2012). These individual catchment outlet runoff time series
were analyzed to map runoff magnitudes and variabilities in time, but also emphasized trends
and spatiotemporal variations, including runoff contributions from the GrIS, the land area
between the GrIS ice margin and the ocean, from the relatively small isolated glaciers and ice
caps, and from entire Greenland. This approach is especially important when trying to
understand the total runoff fraction from Greenland, including the annual and seasonal
freshwater runoff variabilities within individual catchments.

Here, we improve the work by Mernild and Liston (2012) by using an updated version of

SnowModel/HydroFlow and by including a new digital elevation model (DEM). We also extend
the time series to 2014 by using the ERA-Interim (ERA-I) reanalysis products on 3-h time step
(Dee et al. 2011). The objective of this study is to simulate, map, and analyze first-order
atmospheric forcings and GrIS mass balance components for Greenland. The analyzed variables
include the GrIS SMB, together with GrIS surface air temperature, surface melt, precipitation,
evaporation, sublimation, refreezing and retention, and surface freshwater runoff and specific
runoff (runoff volume per time per unit drainage area, $L\ s^{-1}\ km^{-2}$; to convert to $mm\ yr^{-1}$, multiply

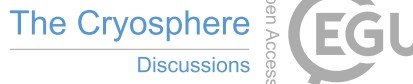



by 31.6) conditions. The time period covers 1979–2014 (35 years), with a focus on the present
day conditions 2005–2014 (the last decade of the simulations). Further, the spatiotemporal
magnitude, distribution, and trends of individual catchment-scale runoff and specific runoff from
Greenland ($n = 3,272$; where $n$ is the number of simulated catchments, each with an individual
flow network) were simulated based on HydroFlow-generated watershed divides and flow
networks for each catchment. The simulated spatiotemporal catchment-scale outlet runoff is
useful as boundary conditions for fjord and ocean model simulations. We also analyzed the
spatiotemporal catchment-scale outlet runoff using Empirical Orthogonal Functions (EOF). This
analysis allowed us to describe simultaneously how the spatial patterns of catchment-scale outlet
runoff changed over time. It also allowed us to explore via cross-correlations the relationship
between the spatiotemporal patterns and large-scale atmospheric-ocean circulation indices
including the North Atlantic Oscillation (NAO) and the Atlantic Multidecadal Oscillation
(AMO), with particular attention to the lag-times, if any, between variations in NAO and AMO
and responses in Greenland catchment-scale runoff.

**2. Model description, setup, and verification**
*2.1 SnowModel*

SnowModel (Liston and Elder 2006a) is established by six sub-models, where five of the

models were used here to quantify spatiotemporal variations in atmospheric forcing, surface
snow properties, GrIS SMB, and Greenland catchment runoff. The sub-model *MicroMet* (Liston
and Elder 2006b; Mernild et al. 2006a) downscaled and distributed the spatiotemporal
atmospheric fields using the Barnes objective interpolation scheme, where the interpolated fields
subsequent were adjusted using known meteorological algorithms, e.g., temperature-elevation,





wind-topography, humidity-cloudiness, and radiation-cloud-topography relationships (Liston and
Elder 2006b). *Enbal* (Liston 1995; Liston et al. 1999) simulated a full surface energy balance
considering the influence of cloud cover, sun angle, topographic slope, and aspect on incoming
solar radiation, and moisture exchanges, e.g., multilayer heat- and mass-transfer processes within
the snow (Liston and Mernild 2012). *SnowTran-3D* (Liston and Sturm 1998, 2002; Liston et al.
2007) accounted for the snow (re)distribution by wind. *SnowPack-ML* (Liston and Mernild 2012)
simulated multilayer snow depths, temperatures, and water-equivalent evolutions. *HydroFlow*
(Liston and Mernild 2012) simulated watershed divides, routing network, flow residence-time,
and runoff routing (configurations based on the hypothetical gridded topography and ocean-mask
datasets), and discharge hydrographs for each grid cell including from catchment outlets. These
sub-models have been tested against independent observations with success in Greenland, Arctic,
high mountain regions, and on the Antarctic Ice Sheet with acceptable results (e.g., Hiemstra et
al. 2006; Liston and Hiemstra 2011; Beamer et al. 2016). For detailed information regarding the
use of SnowModel for the GrIS' or local Greenlandic glaciers' SMB and runoff simulations, we
refer to Mernild and Liston (2010, 2012) and Mernild et al. (2010a, 2014).

*2.2 Meteorological forcing, model configuration and model limitations*

SnowModel was forced with ERA-Interim (ERA-I) reanalysis products on a $0.75°$

longitude $\times 0.75°$ latitude grid from the European Centre for Medium-Range Weather Forecasts
(ECMWF; Dee et al. 2011). The simulations were conducted from 1 September 1979 through 31
August 2014 (35 years) (henceforth 1979–2014), where the 6-hour (precipitation at 12-hour)
temporal resolution ERA-I data was downscaled to 3-hourly values and a 5-km grid using
MicroMet. The 3-hour temporal resolution was chosen to allow SnowModel to resolve the solar



radiation diurnal cycle in its simulation of snow and ice temperature evolution and melt
processes.
The DEM was obtained from Levinsen et al. (2015), and rescaled to a 5-km horizontal
grid increment that covered the GrIS (1,646,175 km$^2$), mountain glaciers, and the entire
Greenland (2,166,725 km$^2$) and the surrounding fjords and seas (Figure 1a). The DEM is time-
invariant specific to the year 2010. The DEM was developed by merging contemporary radar and
laser altimetry data, where radar data were acquired with Envisat and CryoSat-2, and laser data
with the Ice, Cloud, and land Elevation Satellite (ICESat), the Airborne Topographic Mapper
(ATM), and the Land, Vegetation, and Ice Sensor (LVIS). Radar data were corrected for
horizontal, slope-induced, and vertical errors from penetration of the echoes into the subsurface
(Levinsen et al. 2015). Since laser data are not subject to such errors, merging radar and laser
data yields a DEM that resolves both surface depressions and topographic features at higher
altitudes (Levinsen et al. 2015). The distribution of glacier cover was obtained from the
Randolph Glacier Inventory (RGI, v. 5.0) polygons; these data were resampled to the 5-km grid.
The SnowModel land-cover mask defined glaciers to be present when individual grid cells were
covered by 50 % or more of glacier ice.
First, the GrIS DEM was initially divided into six major sections following Rignot et al.
(unpublished): southwest (SW), west (W), northwest (NW), north (N), northeast (NE), and
southwest (SW) (Figure 1b and Table 1). Second, HydroFlow divided Greenland into 3,272
individual catchments (Figure 1c), each with an eight-compass-direction water-flow network
where water is transported through this network via linear reservoirs. Only a single outlet into the
seas was allowed for each individual catchment.



The mean and median catchment sizes were 680 km$^2$ and 75 km$^2$, respectively. The top
one percent of the largest catchments accounted for 53 % of the Greenland area. This distribution
of HydroFlow-defined GrIS catchments (Figure 1c) closely matched both the catchment
distribution by Mernild and Liston (2012) and by Rignot and Kanagaratnam (2006) for the 20
largest GrIS catchments (not including midsize and minor catchments), both with respect to size
and location of the watershed divide. The total number of HydroFlow-generated catchments
presented in this study was ~4 % higher than the number of Greenland catchments in the Mernild
and Liston (2012) study.
In MicroMet, only one-way atmospheric coupling was provided, where the
meteorological conditions were prescribed at each time step. In the natural system, the
atmospheric conditions would be adjusted in response to changes in surface conditions and
properties (Liston and Hiemstra 2011). Due to the use of the 5-km horizontal grid increments,
snow transport and blowing-snow sublimation processes (usually produced by SnowTran-3D in
SnowModel) were excluded from the simulations because blowing snow does not typically move
completely across 5-km distances. Static sublimation was, however, included in the model
integrations. In HydroFlow, the generated catchment divides and flow network were controlled
by the DEM, i.e., exclusively by the surface topography and not by the development of the
glacial drainage system. The role of GrIS bedrock topography on controlling the potentiometric
surface and the associated meltwater flow direction was assumed to be a secondary control on
discharge processes (Cuffey and Paterson 2010).
An example of the HydroFlow generated catchment divides and flow network is
illustrated in detail by Mernild et al. (2017; Figure 1c) for the Kangerlussuaq catchment in
central West Greenland, which includes a part of the GrIS (67°N, 50°W; SW sector of the GrIS).



Because the DEM is time-invariant, no changes though feedbacks from a thinning ice, ice retreat,
and from changes in hypsometry will influence the catchment divides and the flow network
patterns, including the glacial drainage system. Changes in runoff over time are therefore solely
influenced by the climate signal and the surface snow and ice cover conditions (runoff was
generated from gridded inputs from rain, snowmelt, and ice melt), not by the glacial drainage
system. In HydroFlow, the meltwater flow velocities were gained from dye tracer experiments
conducted both through the snowpack (in early and late-summer) and through the englacial and
subglacial environments (Mernild et al. 2006b).

*2.3 Verification*

For Greenland, long-term catchment river runoff observations are sparse; at present

approximately ten permanent hydrometric monitoring stations are operating, measuring the sub-
daily and sub-seasonal runoff variability originating from rain, melting snow, and melting ice
from local glaciers and the GrIS. In addition, these observations only span parts of the runoff
season, ranging between few weeks to approximately three months. For the Kangerlussuaq area,
independent meteorological and snow and ice observational datasets are also available, e.g., K-
transect point observed air temperature and SMB and catchment outlet observed discharge (e.g.,
van de Wal et al. 2005; van den Broeke et al. 2008a; 2008b, Hasholt et. al. 2013). These
observed datasets were used for verification of the SnowModel/HydroFlow ERA-I simulated
GrIS mean annual air temperature (MAAT), GrIS SMB, and catchment freshwater runoff
presented herein (Mernild et al. 2017). These model verifications showed acceptable results (for
further information see Mernild et al. 2017). The use of ERA-I has also showed promising



results after a full evaluation estimating changes in ice sheet surface mass balance for the
catchments linked to Godthåbsfjord (64° N) in Southwest Greenland (Langen et al. 2015).

In the analysis that follows, all correlation trends declared 'significant' are statistically

significant at or above the 5 % level ($p<0.05$; based on a linear regression $t$ test).

*2.4 Surface water balance components*

For the GrIS, surface water balance components can be estimated using the hydrological

method (continuity equation) (Equation 1):

$P - (Su + E) - R + \Delta S = 0 \pm \eta,$                    (1)

where $P$ is precipitation input from snow and rain, $Su$ is sublimation from a static surface, $E$ is
evaporation, $R$ is runoff from snowmelt, ice melt, and rain, $\Delta S$ is change in storage ($\Delta S$ is also
referred to as SMB) derived as the residual value from changes in glacier and snowpack storage.
For snow and ice surfaces, the ablation was estimated as: $Su + E + R$. The amount of snow
refreezing and retention was estimated as: $P_{rain} + melt_{surface} - R$ (for bare ice: $P_{rain} + melt_{surface} =$
$R$). The parameter $\eta$ is the water balance discrepancy. This discrepancy should be 0 (or small), if
the components $P$, $Su$, $E$, $R$, and $\Delta S$ have been determined accurately.

**3. EOF runoff analysis**

We applied an Empirical Orthogonal Function (EOF) analysis to define the

spatiotemporal pattern in simulated catchment outlet runoff. EOF is a statistical tool that
analyzes spatial and temporal runoff data to find combinations of locations that vary consistently

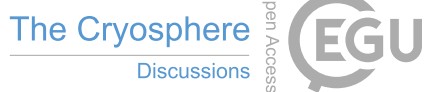

through time, and combinations of time, that vary in a spatially consistent manner (e.g.,
Preisendorfer 1998; Sparnocchia et al. 2003). The major axes of the EOF analysis identify
variations in the catchment outlet runoff in both time and space.
The eigenvalues of the EOFs can be correlated with the temporal data, and the
eigenvectors with spatial locations, to identify how the EOF describes change in runoff in time
and across space. Furthermore, the temporal patterns embedded in the EOFs can, via cross-
correlation analysis, be related to larger scale atmospheric-ocean indices (Mernild et al. 2015), in
this case the North Atlantic Oscillation (NAO) and Atlantic Multi-decadal Oscillation (AMO).
The NAO and AMO indices were obtained from Hurrell and van Loon (1997) and Kaplan et al.
(1998), respectively. This latter analysis can generate hypotheses about whether, for example,
NAO or AMO leads by some years changes in mass balance and runoff (the lag in the cross-
correlation analyses tells us these details).
We focused on the NAO and AMO for several reasons. NAO is estimated based on the
mean sea-level pressure difference between the Azores High and Icelandic Low. NAO is a large-
scale atmospheric circulation index, and is therefore a good measure of airflow and jet-stream
moisture transport variability (e.g., Overland et al. 2012) from the North Atlantic onto Northwest
Europe (Dickson et al. 2000; Rogers et al. 2001). According to Hurrell (1995), a positive NAO is
associated with cold conditions in Greenland, while a negative NAO corresponds to mild
conditions. AMO is a large-scale oceanic circulation index, and an expression of fluctuating
mean sea-surface temperatures in the North Atlantic (Kaplan et al. 1998). For example, Arctic
land surface air temperatures are highly correlated with the AMO (Chylek et al. 2010), and the
overall annual trend in the mean GrIS melt extent correlates with the smoothed trends of the
AMO (Mernild et al. 2011). A positive AMO indicates relatively high surface air temperature



and less precipitation at high latitudes (relatively high net mass balance loss), whereas a negative
AMO indicates relatively low surface air temperature and a higher precipitation (relatively low
net mass balance loss) (Kaplan et al. 1998).

**4. Results and discussion**
*4.1 GrIS surface water balance conditions*

Figure 2 presents the SnowModel ERA-I simulated 35-year mean spatial GrIS surface

MAAT, precipitation, surface melt, evaporation and sublimation, ablation, and SMB. Overall, all
variables follow the expected spatial patterns. For example, the lowest MAAT occurred at the
GrIS interior (≤ -27°C) and highest values were at the margin (≥ 0°C). Also, the lowest annual
mean precipitation values were situated in the northern half of the GrIS interior (≤ 0.25 m water
equivalent (w.e.)), while peak values occurred in the southeastern part of Greenland (≥ 3.5 m
w.e.). The lowest annual mean surface melt values (≤ 0.0625 m w.e.) were present at the upper
parts of the GrIS and vice versa at the lowest margin areas (≥ 5.0 m w.e.). The 35-year mean
SMB illustrated net loss at the lowest elevations of ≥ 4.0 m w.e. and net gain at the highest
elevations of between 0 and 0.25 m w.e. The peak net gain of ≥ 3.5 m w.e. occurred in Southeast
Greenland, which matches what is generally expected from the overall precipitation pattern over
the GrIS. The SnowModel ERA-I spatial simulated 35-year mean distributions generally agree
with previous studies by Fettweis et al. (2008, 2017), Hanna et al. (2011), and Box (2013),
within the different temporal domains covered by these studies.

On GrIS section-scale (Table 1), a clear variability between the six sections occurred for

the surface mass-balance components (Equation 1) for both the 35-year mean and the last
decade. On average, most precipitation fell in the Southeast Greenland sector of 242.6 ± 39.1 Gt



yr$^{-1}$ (where, ± equals one standard deviation). This was likely due to the cyclonicity between
Iceland and Greenland, which typically sets up a prevailing easterly airflow towards the
southeastern coast of Greenland that includes orographic enhancement (Hanna et al. 2006; Bales
et al. 2009). The lowest 35-year mean precipitation of 31.1 ± 5.4 Gt yr$^{-1}$ occurred in the dry
North Greenland. For the last decade, the mean annual precipitation was 232.4 ± 25.2 Gt yr$^{-1}$ and
30.9 ± 5.1 Gt yr$^{-1}$ for Southeast Greenland and North Greenland, respectively. This regional
distribution is in accordance with the study on Greenlandic precipitation patterns by Mernild et
al. (2015), although their analysis was based on observed precipitation from 2001–2012. Further,
in Mernild et al. (2017; Figure 6b), the mean ERA-I grid point precipitation (located closest to
the center of the Kangerlussuaq watershed) was tested against Kangerlussuaq SnowModel ERA-
I downscaled mean catchment precipitation conditions; this analysis indicated no significant
difference between the two datasets.

The ratio between rain and snow precipitation varied from <1 % (Northeast section) to 5

% (Southwest section), averaging 2 % and indicating that rain only played a minor role in the
GrIS precipitation budget (Table 1). For the last decade, the average rainfall-to-snowfall ratio
was 3 % for the entire GrIS.

For the GrIS, the overall precipitation was 653.9 ± 66.4 Gt yr$^{-1}$ (35 years) and 645.0 ±

39.0 Gt yr$^{-1}$ (2005–2014), which is within the lower range of previously reported values
(Fettweis et al. 2017; Table 1). For example, in MAR (Modèle Atmosphérique Régional; v.
3.5.2) the simulated precipitation was between 747.0–642.0 Gt yr$^{-1}$ (1980–1999; snowfall plus
rainfall) forced with a variety of forcings, e.g., ERA-40 (Uppala et al. 2005), ERA-I (Dee et al.
2011), JRA-55 (Japanese 55-year Reanalysis; Kobayashi et al. 2015).



As shown by Fettweis et al. (2017), precipitation is the parameter with the largest
uncertainty due to the spread among the different forcing datasets. Also, systematic observational
errors may occur during precipitation monitoring, such as wind-induced undercatch, because of
turbulence and wind field deformation from the precipitation gauge, wetting losses, and trace
amounts (e.g., Goodison et al. 1989; Metcalfe et al. 1994; Yang et al. 1999; Rasmussen et al.
2012). An understanding of precipitation conditions and uncertainties are therefore highly
relevant for estimating the energy and moisture balances, surface albedo, GrIS SMB conditions,
and, in a broader perspective, the GrIS´s contribution to sea-level changes.
Besides precipitation, melt and ablation are other relevant parameters for estimation and
understanding GrIS surface conditions, where surface melt (including extent, intensity, and
duration) is relevant for SMB conditions. An altered surface melt regime can influence surface
albedo, because wet snow absorbs up to three times more incident solar energy than dry snow
(Steffen 1995), and the energy and moisture balances. Changes in the amount of meltwater also
affect total runoff, ice dynamics, and subglacial lubrication and sliding processes (Hewitt 2013).
Surface melt varied on a section-scale, for the 35-year mean, from $57.2 \pm 24.1$ Gt $yr^{-1}$ in
North Greenland to $155.2 \pm 48.4$ Gt $yr^{-1}$ in Southwest Greenland (Table 1). The average for the
entire GrIS was $542.9 \pm 175.3$ Gt $yr^{-1}$ (Table 1). During the last decade, the surface melt for the
GrIS had increased to $713.4 \pm 138.6$ Gt $yr^{-1}$, varying from $75.9 \pm 26.9$ Gt $yr^{-1}$ in Northeast
Greenland to $202.4 \pm 39.2$ Gt $yr^{-1}$ in Southwest Greenland. This is an increase of 31 % for the
last decade compared to the entire simulation period, which was likely due to increasing MAAT
(assuming an empirical relationship between air temperature (sensible heat) and surface melt
rates) throughout the simulation period (Hanna et al. 2012).

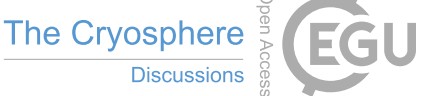



The GrIS ablation patterns varied as expected between the northern and southern sections
from $65.7 \pm 22.6$ Gt yr$^{-1}$ in the north to $132.9 \pm 42.2$ Gt yr$^{-1}$ in the south. For the entire GrIS, the
mean annual ablation was $530.3 \pm 153$ Gt yr$^{-1}$ and $687.8 \pm 118.8$ Gt yr$^{-1}$ for the 35-year period
and 2005–2014, respectively. This was equal to an increase of 30 %, which was also reflected in
the differences in variability from $83.3 \pm 24.7$ Gt yr$^{-1}$ in North Greenland to $175.1 \pm 35.2$ Gt yr$^{-1}$
in Southwest Greenland (Table 1).
Runoff is a part of the ablation budget and therefore must be quantified to understand
GrIS mass balance changes. Runoff varied from $50.0 \pm 22.7$ Gt yr$^{-1}$ in North Greenland to 112.6
$\pm 41.8$ Gt yr$^{-1}$ in South Greenland, averaging $418.1 \pm 151.1$ Gt yr$^{-1}$ for the 35-year mean period.
For 2005–2014, the mean runoff was $73.7 \pm 119.8$ Gt yr$^{-1}$; a 37 % increase (Table 1). This
increase confirms the results from previous studies (e.g., Wilton et al. 2016). On a regional-scale,
runoff varied from $67.6 \pm 25.0$ Gt yr$^{-1}$ in North Greenland to $154.4 \pm 36.3$ Gt yr$^{-1}$ in Southwest
Greenland. The simulated section runoff distribution was largely in agreement with trends noted
by Lewis and Smith (2009) and Mernild and Liston (2012). The section runoff variability
roughly followed the precipitation patterns, where sections with high precipitation equaled low
runoff (e.g., in Southeast Greenland) and vice versa (e.g., in Southwest Greenland). More
specifically, GrIS snowpack retention and refreezing processes suggest that sections with
relatively high surface runoff were synchronous with relatively low end-of-winter snow
accumulation because more meltwater was retained in the thicker, colder snowpack, reducing
and delaying runoff to the internal glacier drainage system (e.g., Hanna et al. 2008). However, in
maritime regions such as Southeast Greenland, high surface runoff can result from abnormally
wet conditions (Mernild et al. 2014). Furthermore, runoff was negatively correlated to surface
albedo and snow cold content, as confirmed by Hanna et al. (2008) and Ettema et al. (2009).



For the dry North and Northeast Greenland (Table 1), the relatively low end-of-winter
snowpack melted relatively fast during spring warm-up. After the winter snowpack had ablated,
the ice surface albedo promoted a stronger radiation-driven ablation and surface runoff, owing to
the lower ice albedo. For the wetter Southeast Greenland (Table 1), the relatively high end-of-
winter snow accumulation, combined with frequent summer snow precipitation events, kept the
albedo high. Therefore, in that region it generally took longer time to melt the snowpack
compared to the drier parts of the GrIS before ablation started to affect the underlying glacier ice.
Regarding specific runoff (runoff volume per unit drainage area per time, L s$^{-1}$ km$^{-2}$; to
convert to mm yr$^{-1}$, multiply by 31.6), maximum values of 16.7 L s$^{-1}$ km$^{-2}$ and 22.9 L s$^{-1}$ km$^{-2}$
were seen in Southwest Greenland for the mean 35-year and 2005–2014 periods, respectively.
The minimum values of 4.4 L s$^{-1}$ km$^{-2}$ and 6.2 L s$^{-1}$ km$^{-2}$ for the mean 35-year and 2005–2014
periods, respectively, occurred in Northeast Greenland (Table 2). On average for the GrIS, the
corresponding specific runoffs were 8.1 L s$^{-1}$ km$^{-2}$ and 11.1 L s$^{-1}$ km$^{-2}$, respectively, which are
within the range of previous studies (e.g., Mernild et al. 2008). Specific runoff is a valuable tool
for comparing runoff on regional and catchment scales, and it can also be used to quantifying the
absolute runoff contributions from increasing runoff and increasing melt area extent. The
difference in specific runoff between the two periods indicates that the increase in runoff has
increased faster than the increase in melt area extent.
Refreezing and retention in the snow and firn packs were defined as rain plus surface
melt minus runoff (see Section 2.4). For the GrIS, the 35-year mean refreezing and retention was
estimated to be 25 % (140.1 ± 35.5 Gt yr$^{-1}$), and it was 22 % (158.4 ± 34.4 Gt yr$^{-1}$) for 2005–
2014 (Table 1). Hence, refreezing and retention provided an important quantitative contribution
to the evolution of snow and firn layers, ice densities, snow temperatures (cold content or snow



temperatures below freezing), and moisture available for runoff (Liston and Mernild 2012). The
SnowModel ERA-I refreezing and retention simulations were within the order of magnitude
produced by the single-layer snowpack model used by Hanna et al. (2008), but lower than the 45
% simulated by Ettema et al. (2009). On the regional-scale, the 35-year mean refreezing and
retention value varied from 13 % in North Greenland to 30 % in both Southeast and Southwest
Greenland. For 2005–2014, the values were 12 % for North Greenland and 32 % for Southeast
Greenland (Table 1), indicating a clear variability in refreezing and retention between the
different regions.

In Figure 3a, the time series of GrIS mean annual refreezing and retention shows an

increasing trend (significant) and variability ranging from ~0.05 m w.e. (1992) to ~0.14 m w.e.
(2012), with an annual mean value of 0.09 ± 0.02 m w.e. In Figure 3b, the spatial 35-year mean
GrIS refreezing and retention is presented together with values from 1992 and 2012, the
minimum and maximum years, respectively. The mean spatial distribution highlights minimal
refreezing and retention at the GrIS interior, whereas areas with low elevation had values above
0.8 m w.e. in southern part of the GrIS. For the minimum year 1992, the pattern was more
pronounced with no refreezing and retention in the interior. The maximum year 2012 on the
other hand had refreezing and retention at the interior (between 0 and 0.02 m w.e.) (Figure 3b).
This was likely due to the extreme GrIS surface melt event throughout July 2012 (e.g., Nghiem
et al. 2012; Hanna et al. 2014). When divided into regions and catchments, the 2012 simulated
refreezing and retention showed a clear separation between highest values in Southwest
Greenland and lowest values in Northeast and East Greenland. Because here, refreezing and
retention were estimated as the sum of rain and melt minus the sum of runoff, this SnowModel
analysis did not provide a detailed description of the physical mechanisms and conditions





(beyond the standard SnowModel snowpack temperature and density evolution) leading to, e.g.,
non-linearities in snow and firn meltwater retention (Brown et al. 2012). However, while likely
an oversimplification of the natural system, this quantitative estimation of refreezing and
retention is an important step forward, and improves our runoff and the associated SMB
estimates. A model that does not include refreezing and retention processes in its snow and firn
evolution calculations, and the associated impacts on SMB, will introduce additional uncertainty
in it calculations of GrIS SMB and its contribution to sea-level change.

The GrIS SMB for the 35-year mean was $123.7 \pm 163.2$ Gt $yr^{-1}$, indicating a negative sea-

level contribution, and $-42.9 \pm 133.5$ Gt $yr^{-1}$ for 2005–2014, indicating a trend towards a positive
sea-level contribution (Table 1). This change in SMB between the two periods was mainly due to
an increase in runoff of 155.6 Gt $yr^{-1}$, where other water balance components showed relatively
lesser increases. For the GrIS, the 35-year mean SMB was negative for the northern regions,
positive for the southern regions and only positive for the southeastern sector for 2005–2014.
Overall, the SMB patterns were highly controlled by the distribution of precipitation and runoff.

The linear trends for the different water balance components are shown in Table 1. For

the 35-year period, only significant trends occurred for rain, surface melt, runoff, ablation, and
SMB (highlighted in bold in Table 1), where all except SMB showed positive trends (note that
SMB loss is calculated as negative by convention). In Figure 4, selected GrIS parameters are
illustrated, where, for example, SMB showed a negative trend of $-99.2$ Gt $decade^{-1}$ (significant),
heading towards a zero balance at the end of the simulation period (Figure 4). For 2005–2014,
however, the SMB trend was positive 24.2 Gt $decade^{-1}$ (insignificant). Similar positive SMB
trends have previously been shown in studies by Hanna et al. (2011), Tedesco et al. (2014),
Fettweis et al., (2008, 2011, 2013) and Wilton et al. (2016), even though variabilities in mean



SMB occur between the different studies. Wilton et al. (2016) estimated the GrIS SMB to be
~100 Gt yr$^{-1}$ in the late-2000s. Further, for 2005–2014, air temperature, precipitation, surface
melt, sublimation and evaporation, and runoff trends were all negative (insignificant) (Figure 4
and Table 1).

*4.2 Greenland spatiotemporal runoff distribution and EOF analysis*

The Greenland 35-year simulated catchment outlet runoff and specific runoff distribution

are shown in Figure 5. Each circle represents the volume (individual catchment outlet
hydrographs are not shown), including runoff from thousands of glaciers located between the
GrIS margin and the surrounding seas. The 35-year mean catchment outlet runoff varied from
<0.0001 to 25.7 × 10$^9$ m$^3$ (Figure 5a) and specific runoff from <0.1 to 127.5 L s$^{-1}$ km$^{-2}$ (Figure
5b). Catchment runoff variability depends on the regional climate conditions, land-ice area
cover, elevation range (including hypsometry) within each catchment, and catchment area. Here
the length in runoff season varied from two to three weeks in the north to four to six months in
the south. The median annual catchment runoff and specific runoff were 0.025 × 10$^9$ m$^3$ and 9.1
L s$^{-1}$ km$^{-2}$, respectively. The median specific runoff value is in agreement with previous studies
(e.g., Mernild et al. 2010a). Further, the variance in catchment runoff and specific runoff varied
from <0.0001 to 8.3 × 10$^9$ m$^3$ and <0.01 to 19.3 L s$^{-1}$ km$^{-2}$, respectively, with a median variance
of 0.006 × 10$^9$ m$^3$ and 2.4 L s$^{-1}$ km$^{-2}$ (Figures 4a and 4b). Regarding the linear trend in annual
runoff, both increasing and decreasing trends occurred over the 35 years. In total, 81 % (19 %) of
all catchments had increasing (decreasing) runoff trends over the 35 years (all of the decreasing
trends were insignificant). For western Greenland catchments, only increasing runoff trends
occurred (Figures 4a and 4b). The runoff and specific runoff trends varied among catchments



from -0.09 to 5.4 × 10$^9$ m$^3$ decade$^{-1}$ and from -1.3 to 12.9 L s$^{-1}$ km$^{-2}$ decade$^{-1}$, respectively, with a
median value of 0.001 × 10$^9$ m$^3$ and 0.5 L s$^{-1}$ km$^{-2}$ decade$^{-1}$ (Figures 4a and 4b).

The EOF analysis of runoff returned three axes that captured 25, 17 and 12 % of the

variation in runoff from the simulated SnowModel ERA-I annual catchment runoff (Figure 6).
Following several significance tests, only EOF1 captured significant variation. In Figure 6, the
temporal pattern in EOF1, with a 5-year running mean, reveals a pattern of positive running
mean values for the first two decades of the simulation period (1979–1999), and negative values
hereafter (2000–2014). When EOF1 is positive, Greenland runoff is relatively low and vice versa
(Figure 7). Overall, this indicates a positive temporal trend in runoff; as EOF1 goes down, runoff
goes up. While not significant based on EOF test metrics, EOF2 and EOF3 patterns are less
pronounced and in anti-phase to each other (Figure 6).

The temporal cycle of EOF patterns has associated spatial elements, derived from the

eigenvectors (Figure 8). The eigenvectors in Figure 8 reveal the spatial pattern as a correlation
between temporal trends captured by the EOFs and each individual Greenland catchment. These
data indicate that the temporal trend of increasing runoff captured in EOF1 is shared by nearly all
catchments in Greenland. Because decreasing EOF1 values indicate increasing runoff, a negative
correlation with EOF1 in space indicates increasing runoff. Catchment numbers greater than
#2500 (Figure 8) are located in Southeast Greenland and are in contrast to this. These catchments
experience a distinct out-of-phase pattern of runoff compared to the overall Greenland conditions
for the last 35 years.

This difference between Southeast Greenland and the rest of Greenland supports previous

findings (e.g., Lenaerts et al. 2015) proposing that variabilities in runoff are not only influenced
by melt conditions, but also by precipitation patterns (primarily the end-of-winter snow



accumulation), where high precipitation equals low runoff conditions such as in Southeast
Greenland. Furthermore, patterns were also detected to be associated to EOF2 and EOF3
(Figures 8b and 8c). These EOF2 and EOF3 patterns differed from EOF1, and they were
associated with a different geographic breakdown, where both positive and negative correlations
were seen for all regions. The physical mechanism behind these distributions is not clear.
There were strong correlations between the EOF1 and regional climate patterns expressed
by the AMO and NAO (Figure 9). We found a negative correlation between EOF1 and AMO ($r$
$= 0.68$; significant, $p<0.01$), suggesting that stronger AMO is associated with lower EOF1 values
which are indicative of higher runoff (Figure 9a). In contrast, we found a positive correlation
between EOF1 and NAO ($r = 0.40$; significant, $p<0.01$), suggesting that stronger NAO values
are associated with higher EOF1 values which are indicative of lower runoff (Figure 9b).
Additional insight into the time frame over which these correlations arises is seen in Figure 9.
For AMO, the lags are centered near zero, suggesting an immediate, real time correlation
between AMO and runoff. In contrast, the strongest lag in the NAO-EOF1 relationships is at -2,
suggesting a short delay in effects. Lags of 0 and -2 are not large, indicating that overall, large-
scale natural variability in AMO and NAO are closely associated in time to catchment runoff
variations in Greenland.
Mernild et al. (2011) emphasized that trends in AMO (smoothed) was analogous to trends
in GrIS melt extent, where increasing AMO equaled increasing melt extent, and vice versa.
Further, Chylek et al. (2010) showed that the Arctic detrended temperatures were highly
correlated with AMO. However, this issue requires further investigation to establish the details
of, and the mechanisms behind, the interrelationships.



## 5. Conclusions

Greenland catchment outlet runoff is rarely observed and studied, although quantification of runoff from Greenland is crucial for our understanding of the link between a changing climate and changes in the cryosphere, hydrosphere, and atmosphere. We have reconstructed the impact of changes in climate conditions on hydrological processes at the surface of the GrIS for the 35-year period 1979–2014. We have also simulated the Greenland spatiotemporal distribution of refreezing and retention, and freshwater runoff to surrounding seas by merging SnowModel (a spatially distributed meteorological, full surface energy balance, snow and ice evolution model) with HydroFlow (a linear-reservoir run-off routing model) forced by ERA-I atmospheric forcing data. Before simulating the individual catchment runoff to downstream areas, the catchment divides and flow networks were estimated, yielding a total of 3,272 catchments in Greenland.

For the GrIS, the simulated spatial distribution and time series of surface hydrological processes were in accordance with previous studies, although precipitation and SMB were in the lower range of these studies. Overall, Greenland has warmed and the runoff from Greenland has increased in magnitude. Specifically, 81 % of the catchments showed increasing runoff trends over the simulation period, with relatively high and low mean catchment runoff from the southwestern and northern parts of Greenland, respectively. This indicates distinct regional-scale runoff variability in Greenland. Runoff variability with near zero lag time suggests a real-time covariation between the pattern in EOF1 and changes in AMO and NAO. This indicates that large-scale natural variability in AMO and NAO is closely related to catchment runoff variations in Greenland. The physical mechanism behind this phenomenon is unclear, unless it is a response to "long-term" cycles in AMO and NAO.





The simulated runoff can be used as boundary conditions in ocean models to understand
hydrologic links between terrestrial and marine environments in the Arctic. Changes and
variability in runoff from Greenland are expected to play an essential role in the hydrographic
and circulation conditions in fjords and the surrounding ocean under a changing climate.

**Acknowledgements**
We thank the Japan Society for the Promotion of Science (JSPS) for financial support under
project number S17096, and the Western Norway University of Applied Sciences (HVL) for
travel funds. All model data requests should be addressed to the first author. The authors have no
conflict of interest.



















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



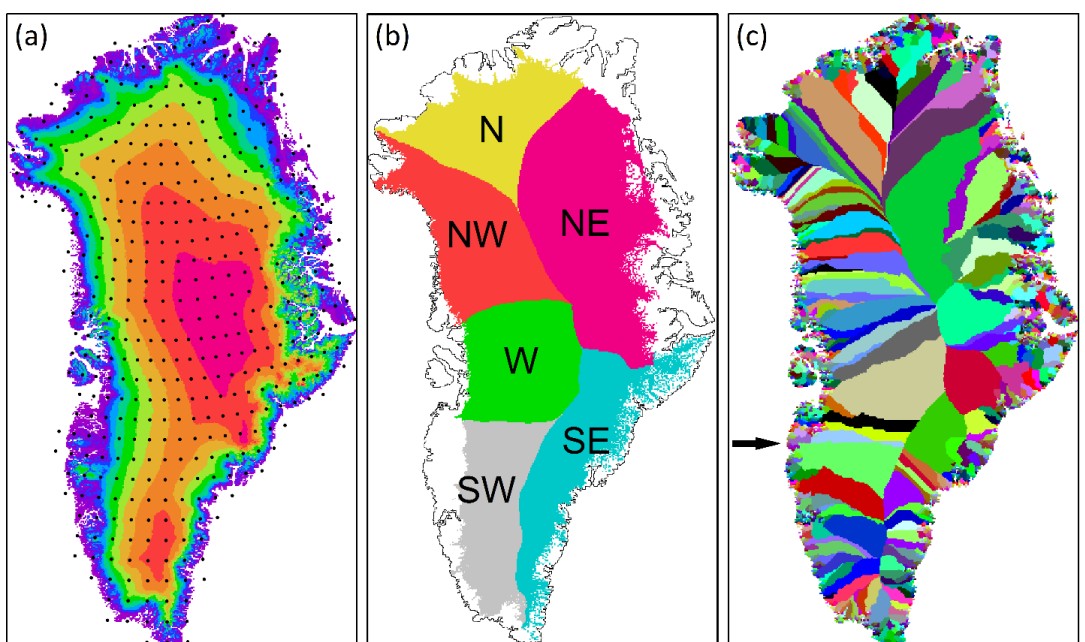


**Figure 1**: (a) Greenland simulation domain with topography (500-m contour interval) and
locations of ERA-I atmospheric forcing grid points used in the model simulations (black dots; to
improve clarity only every other grid point was plotted in x and y, i.e., 25 % of the grid points
used are shown); (b) the major regional division of the GrIS following Rignot et al.
(unpublished); and (c) HydroFlow simulated individual Greenland drainage catchments ($n =$
3,272; represented by multiple colors). The approximate location of the Kangerlussuaq
catchment is shown with a black arrow.









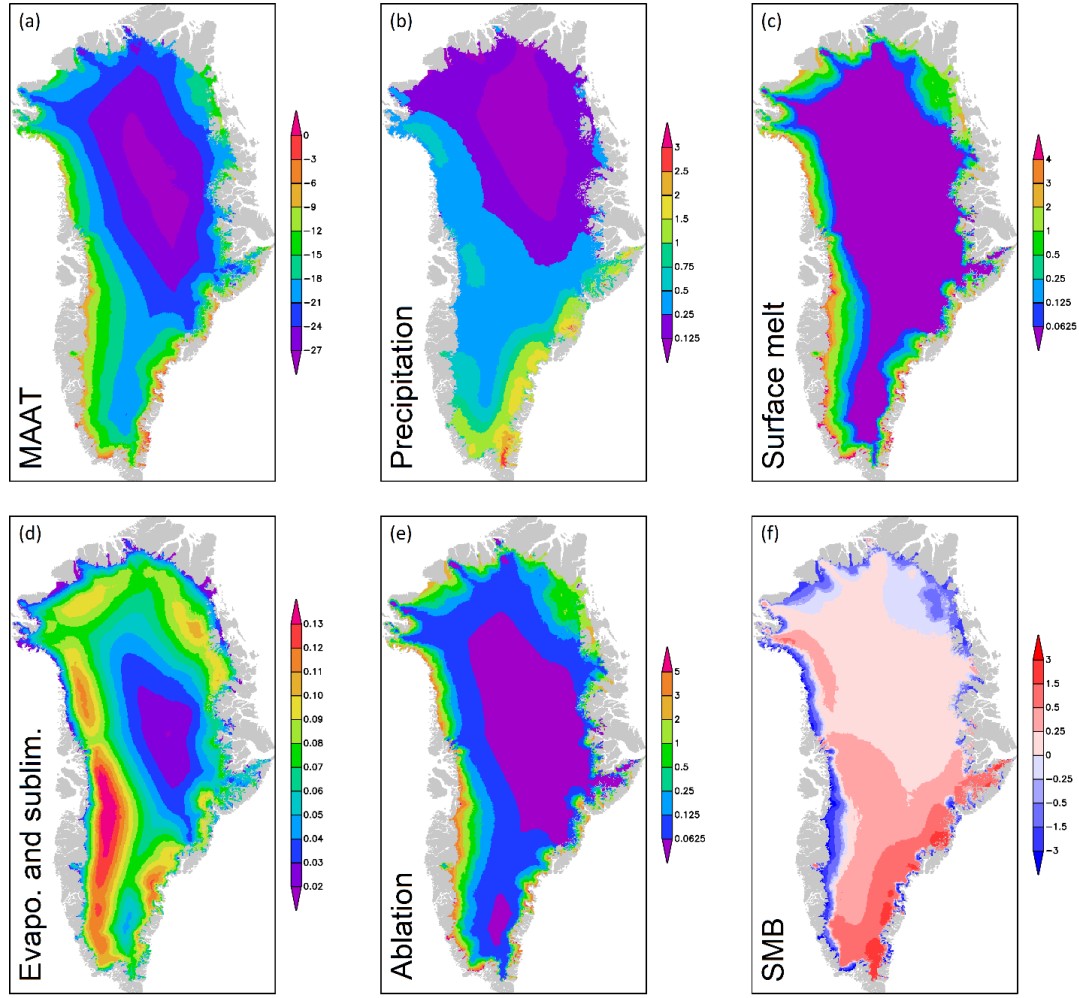


**Figure 2:** SnowModel ERA-I simulated 35-year mean spatial GrIS surface (1979–2014): (a)
MAAT (°C); (b) precipitation (m w.e.); (c) surface melt (snow and ice melt) (m w.e.); (d)
evaporation and sublimation (m w.e.); (e) ablation (m w.e.); and (f) SMB (m w.e.).











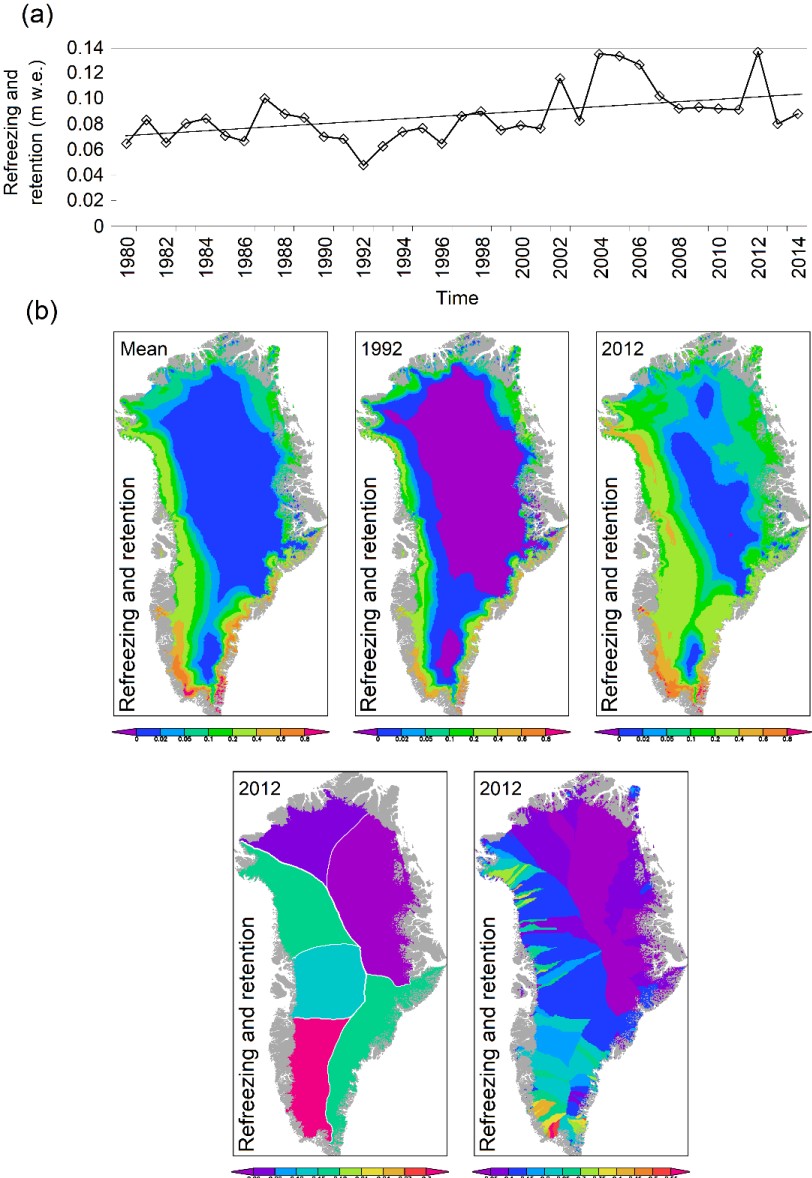


**Figure 3:** (a) SnowModel ERA-I simulated time series of GrIS mean annual refreezing and
retention (1979–2014) (m w.e.); and (b) spatial 35-year mean GrIS refreezing and retention and
annual values (m w.e) for 1992 and 2012 (upper row), together with the 2012-division into
regions (lower row left) and catchments (lower row right).



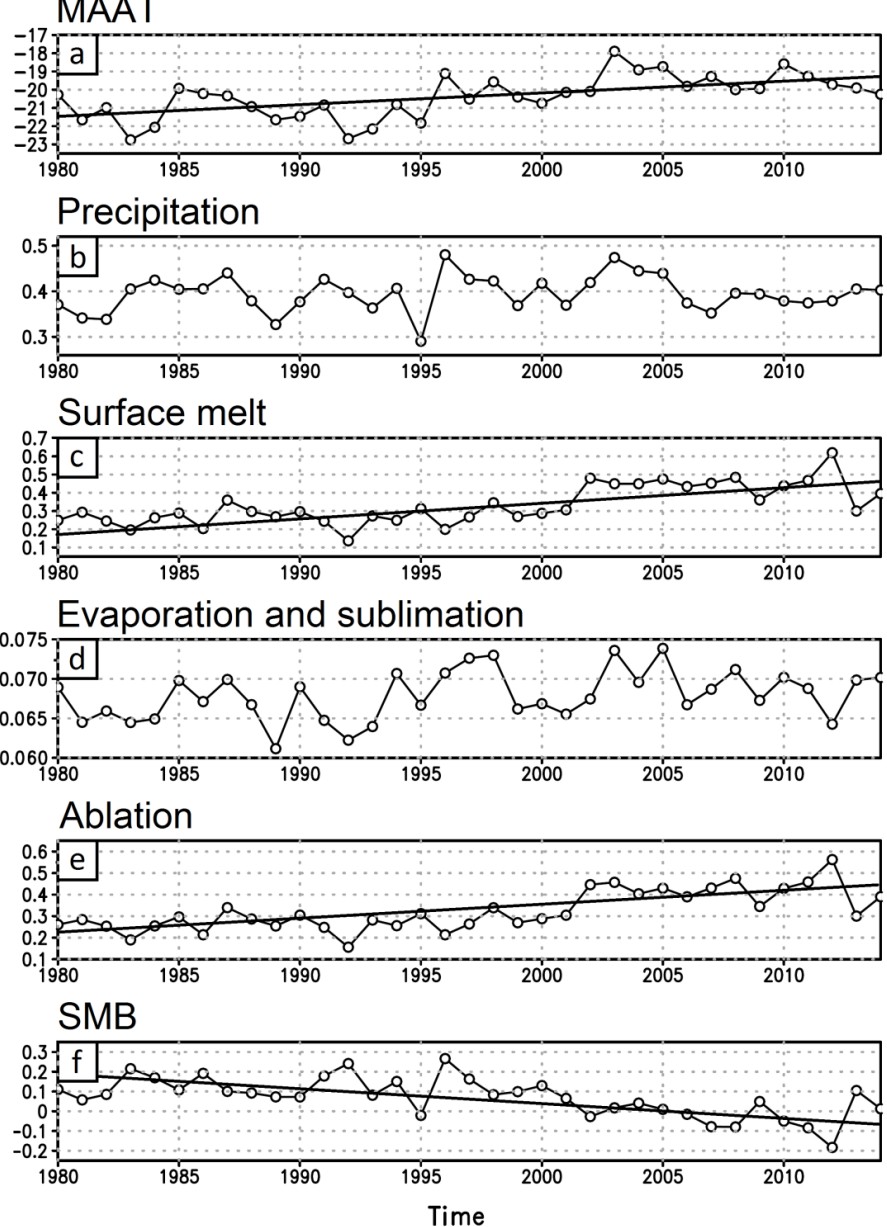


**Figure 4:** SnowModel ERA-I simulated time series of GrIS annual mean (1979–2014): (a)
MAAT (°C); (b) precipitation (m w.e.); (c) surface melt (snow and ice melt) (m w.e.); (d)
evaporation and sublimation (m w.e.); (e) ablation (m w.e.); and (f) SMB (m w.e.). Only
significant linear trends are shown.



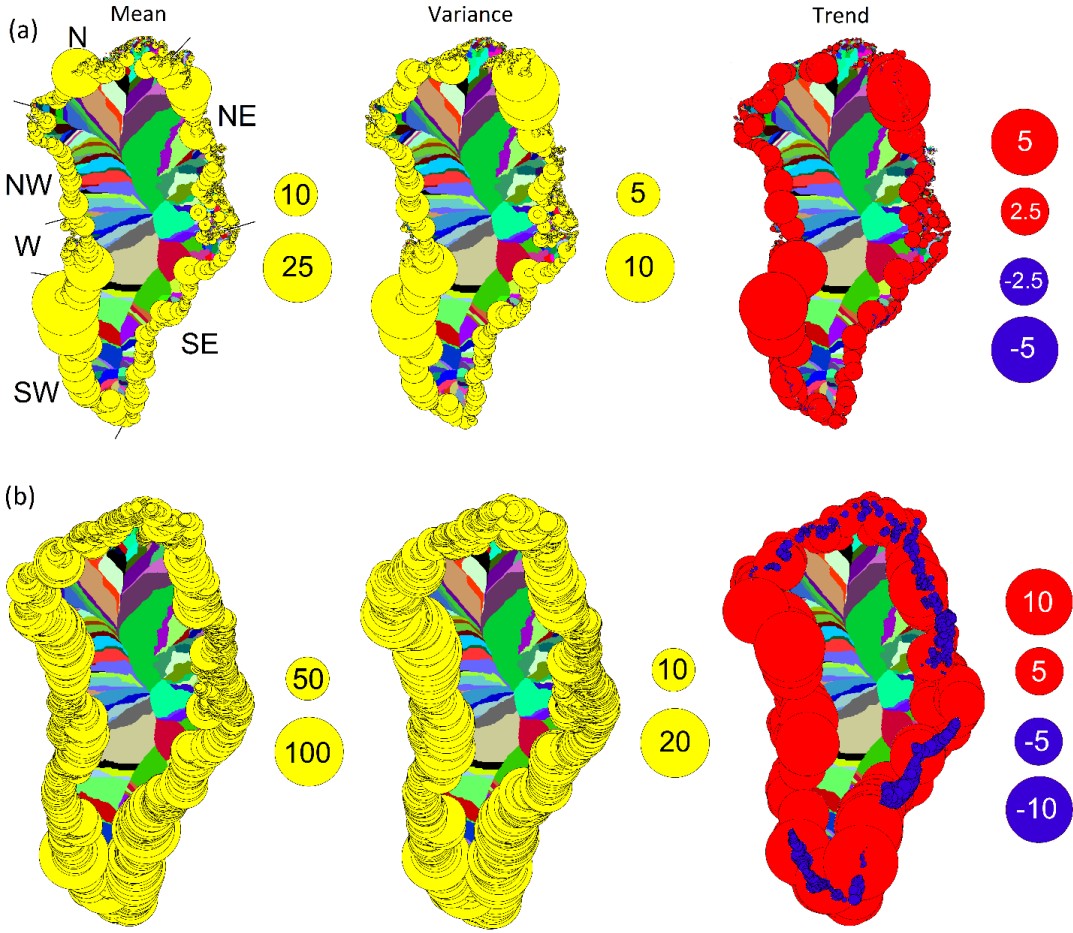

**Figure 5:** SnowModel ERA-I simulated 35-year spatial Greenland catchment runoff (1979–
2014): (a) mean runoff ($\times 10^9$ m$^3$) (the locations of the major regions SW, W, NW, etc., are
illustrated), runoff variance (here illustrated as one standard deviation; $\times 10^9$ m$^3$), and decadal
runoff trends (linear; $\times 10^9$ m$^3$ decade$^{-1}$) (catchments with increasing runoff trends are shown
with red and decreasing trends with blue colors); and (b) mean specific runoff (L s$^{-1}$ km$^{-2}$),
specific runoff variance (L s$^{-1}$ km$^{-2}$), and specific runoff trends (linear; L s$^{-1}$ km$^{-2}$ decade$^{-1}$).





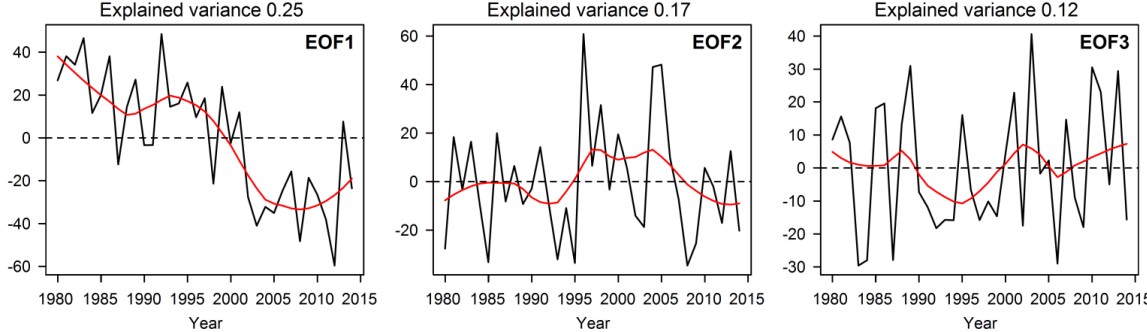

**Figure 6:** SnowModel ERA-I simulated runoff time series (1979–2014) of the empirical

orthogonal functions (black curve) and 5-year running mean smoothing line (red curve) of EOF1,

EOF2, and EOF3. The explained variance is shown for each EOF, where only EOF1 is

significant.



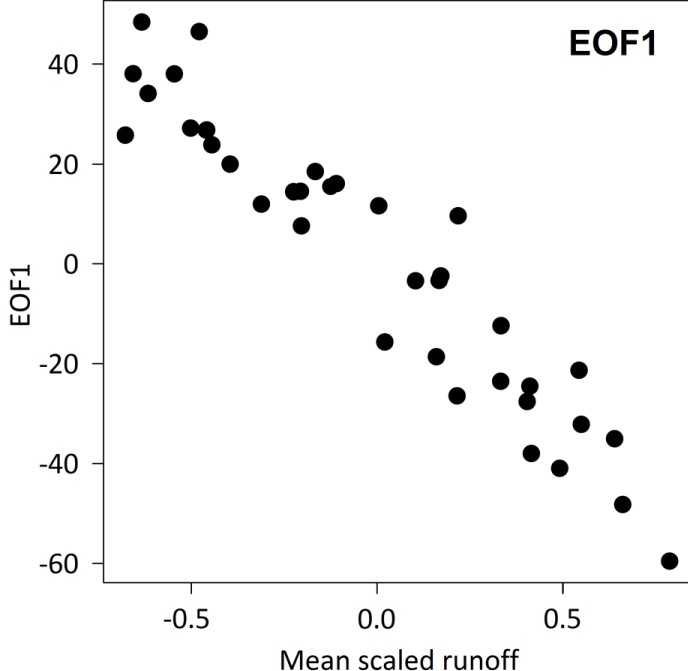

**Figure 7:** EOF1 cross correlation relationships with mean annual scaled runoff from Greenland.



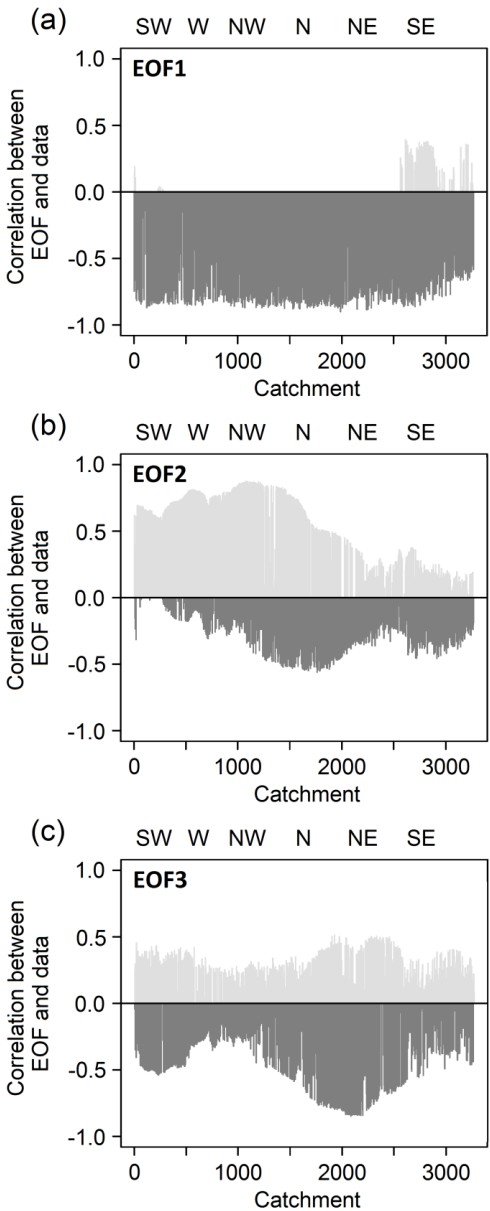

1070

**Figure 8:** Eigenvector correlation values for each simulated catchment (1 to 3,272) for: (a)

EOF1; (b) EOF2; and (c) EOF3. From left to right on the lower x-axis the catchments follows the

clockwise path from the southern tip of Greenland (Southwest Greenland, Catchment 1) to the

northern part (N section) and back to the southern tip (Southeast Greenland, Catchment 3,272).

The location of the major regions: SW, W, NW, etc., are shown on the upper x-axis.





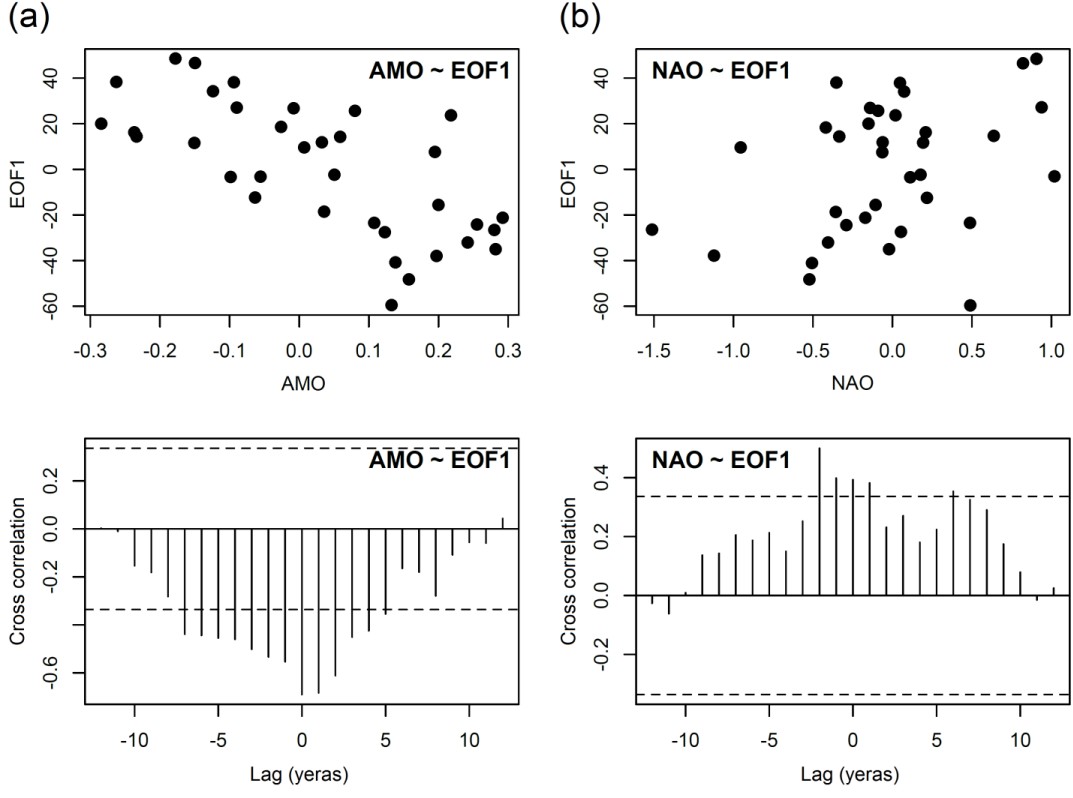


**Figure 9:** EOF1 cross correlation relationships between simulated Greenland runoff: (a) AMO

and (b) NAO. The horizontal dashed lines on each of the column charts indicate the significance

(95% confidence).











**Table 1:** Regional breakdown of GrIS surface mean annual conditions (units are in Gt) and trends (linear; Gt decade$^{-1}$): precipitation (P) (including rain and snow), surface melt, evaporation (E) and sublimation (Su), runoff (R), ablation, refreezing and retention, and surface mass-balance (SMB) for GrIS and for each of the six regions both from 1979–2014 (35 years) and 2005–2014 (10 years). Specifically for rain the %-value of total precipitation is shown. Trends are shown in paragraphs for the GrIS column. Significant trends ($p < 0.05$) are highlighted in bold.

| | N (229,075 km²) | NE (454,900 km²) | SE (250,425 km²) | SW (213,550 km²) | W (231,150 km²) | NW (267,075 km²) | GrIS (1,646,175 km²) |
|---|---|---|---|---|---|---|---|
| 1979–2014 | | | | | | | |
| P | 31.1 ± 5.4 | 68.4 ± 10.6 | 242.6 ± 39.1 | 142.3 ± 23.3 | 85.4 ± 13.5 | 84.2 ± 13.9 | 653.9 ± 66.4 (9.0) |
| P (rain) | 0.4 ± 0.2 (1 %) | 0.4 ± 0.2 (<1 %) | 4.2 ± 1.8 (2 %) | 6.6 ± 2.8 (5 %) | 1.7 ± 0.8 (2 %) | 2.2 ± 1.1 (3 %) | 15.3 ± 5.4 (2 %) (**3.0**) |
| P (snow) | 30.7 ± 5.3 | 68.0 ± 10.5 | 238.5 ± 38.7 | 135.7 ± 22.7 | 83.7 ± 13.2 | 82.0 ± 13.5 | 638.6 ± 65.0 (6.0) |
| Surface melt | 57.2 ± 24.1 | 72.2 ± 33.8 | 101.0 ± 27.1 | 155.2 ± 48.4 | 67.4 ± 24.0 | 89.8 ± 33.2 | 542.9 ± 175.3 (**121.7**) |
| E + Su | 15.7 ± 0.9 | 25.3 ± 1.6 | 16.8 ± 0.8 | 20.3 ± 1.7 | 16.4 ± 1.1 | 17.7 ± 0.9 | 112.2 ± 5.2 (1.8) |
| R | 50.0 ± 22.7 | 62.6 ± 31.5 | 69.2 ± 21.0 | 112.6 ± 41.8 | 53.6 ± 19.4 | 70.0 ± 28.1 | 418.1 ± 151.1 (**106.4**) |
| Ablation (E + Su + R) | 65.7 ± 22.6 | 87.9 ± 31.5 | 86.0 ± 21.4 | 132.9 ± 42.2 | 70.0 ± 19.9 | 87.7 ± 28.5 | 530.3 ± 153.0 (**108.2**) |
| Refreezing and retention (rain and surface melt minus runoff) | 7.6 ± 2.8 (13 %) | 10.0 ± 4.0 (14 %) | 36.0 ± 8.9 (30 %) | 49.2 ± 13.3 (30 %) | 15.5 ± 6.1 (22 %) | 22.0 ± 7.2 (24 %) | 140.1 ± 35.5 (25 %) (**18.3**) |
| SMB | -34.6 ± 24.8 | -19.6 ± 32.4 | 156.6 ± 44.5 | 9.3 ± 50.3 | 15.4 ± 23.6 | -3.5 ± 32.1 | 123.7 ± 163.2 (**-99.2**) |
| 2005–2014 | | | | | | | |
| P | 30.9 ± 5.1 | 71.0 ± 11.9 | 232.4 ± 25.2 | 138.5 ± 16.1 | 86.4 ± 8.6 | 85.3 ± 16.9 | 645.0 ± 39.0 (-5.1) |
| P (rain) | 0.5 ± 0.3 (2 %) | 0.4 ± 0.2 (<1 %) | 5.2 ± 1.9 (2 %) | 7.8 ± 2.3 (6 %) | 2.0 ± 0.6 (2 %) | 2.9 ± 1.3 (4 %) | 18.7 ± 3.4 (3 %) (-2.8) |
| P (snow) | 30.4 ± 5.0 | 70.6 ± 11.9 | 227.1 ± 25.0 | 130.8 ± 15.7 | 84.4 ± 8.6 | 82.9 ± 16.4 | 626.3 ± 39.2 (-2.3) |
| Surface melt | 75.9 ± 26.9 | 101.7 ± 34.5 | 129.7 ± 16.3 | 202.4 ± 39.2 | 89.3 ± 19.7 | 124.6 ± 26.8 | 713.4 ± 138.6 (-79.7) |
| E + Su | 15.7 ± 1.0 | 25.9 ± 1.1 | 17.3 ± 0.9 | 20.7 ± 1.5 | 16.7 ± 0.8 | 17.8 ± 0.9 | 114.1 ± 4.3 (-3.8) |
| R | 67.6 ± 25.0 | 89.3 ± 31.0 | 91.7 ± 14.4 | 154.4 ± 36.3 | 71.2 ± 15.2 | 99.5 ± 22.4 | 573.7 ± 119.8 (-26.0) |
| Ablation (E + Su + R) | 83.3 ± 24.7 | 115.2 ± 30.8 | 109.0 ± 14.2 | 175.1 ± 35.2 | 87.9 ± 15.0 | 117.3 ± 22.5 | 687.8 ± 118.8 (-29.8) |
| Refreezing and retention (rain and surface melt minus runoff) | 8.8 ± 3.5 (12 %) | 12.8 ± 5.3 (13 %) | 43.3 ± 6.9 (32 %) | 55.8 ± 12.1 (27 %) | 20.1 ± 6.0 (22 %) | 28.0 ± 8.1 (22 %) | 158.4 ± 34.4 (22 %) (-56.6) |
| SMB | -52.4 ± 26.3 | -44.2 ± 30.6 | 123.4 ± 35.7 | -36.6 ± 45.0 | -1.5 ± 16.4 | -31.5 ± 29.7 | -42.9 ± 133.5 (24.7) |





**Table 2:** Regional breakdown of GrIS specific runoff (L s$^{-1}$ km$^{-2}$) and changes in specific runoff
(linear; L s$^{-1}$ km$^{-2}$ decade$^{-1}$) for GrIS and each of the six individual sections both from 1979–
2014 and 2005–2014. Changes in specific runoff are shown in the brackets.

| | N | NE | SE | SW | W | NW | GrIS |
|---|---|---|---|---|---|---|---|
| 1979–2014 | 6.9 (1.6) | 4.4 (1.5) | 8.8 (1.9) | 16.7 (4.1) | 7.4 (1.8) | 8.3 (2.1) | 8.1 (2.0) |
| 2005–2014 | 9.4 (-0.5) | 6.2 (-1.0) | 11.6 (-0.1) | 22.9 (0.7) | 9.8 (-0.5) | 11.8 (1.5) | 11.1 (-1.3) |
