# Peer review of "mass balance and the spatiotemporal distribution of"

_The Cryosphere, 2017_

## Referee Comment (RC1) · Anonymous Referee #1 · 12 Jan 2018

**Review of: "Reconstruction of the Greenland Ice Sheet surface mass balance and spatiotemporal distribution of freshwater runoff from Greenland to surrounding seas"**, by *S. H. Mernild et al.*, submitted to *The Cryosphere*.

**General comments**

Using the updated version of SnowModel/HydroFlow, the authors simulate the surface mass balance (SMB) and components, i.e. runoff, melt and retention, of the Greenland ice sheet (GrIS) at 5 km resolution for the period 1979-2014. Precipitation is downscaled from ERA-Interim re-analysis for the same period. The model includes a snow module accounting for meltwater retention in snow, an energy balance scheme and a meltwater runoff routing module. This routing module allows to quantify the runoff contribution of 6 GrIS sectors, further refined to over 3,000 individual catchments. The authors first discuss the modeled contemporary (1979-2014) GrIS SMB and recent trends (2005-2014) in these 6 sectors. The analysis is further extended to the numerous individual catchments to show that about 80% of these have experienced increasing meltwater runoff since 1979. Then, the authors correlate this recent runoff increase with the natural variability of the Atlantic Multidecadal Oscillation (AMO) and North Atlantic Oscillation (NAO).

The manuscript is overall well written, but the results presented seem inaccurate and raise questions and concerns on the model ability to reproduce the contemporary GrIS SMB, potentially altering the conclusions drawn in this paper. These concerns are summarized in the Substantive Comments. In brief, I suspect that the current version of SnowModel/HydroFlow has several issues resulting in inaccurate SMB estimates over the GrIS. Compared to other studies, runoff is significantly overestimated likely due to inaccurate representation of meltwater retention in firn. This study suggests that only 25% of melt is retained in the firn pack while most recent efforts demonstrated that it is closer to ~45%. Such a retention underestimation has severe implications on runoff calculation and SMB. For instance, the authors present negative modeled GrIS-integrated SMB for the period 2005-2014 as opposed to recently published GrIS mass balance and GRACE studies. This makes these results potentially unreliable and the conclusions drawn inconsistent.

For these reasons, I judge that the manuscript **cannot be published** in the current form and needs **major revisions**, unless the authors prove that their estimates of precipitation, runoff, melt, refreezing and SMB are reasonably accurate. To achieve this, the authors **must** perform a thorough model evaluation against in situ SMB measurements and compare their results to remote sensing mass change records, and mass balance estimates compiled in previously published studies. This would highlight potential issues in their snow model and provide some insight on how to solve them. If these evaluations/comparisons and validation of the model results can be successfully achieved, and if the suggested corrections listed hereunder are applied, I would be happy to reassess this manuscript.

**Substantive Comments**

1. Throughout the manuscript, the authors discuss changes in SMB components and recent trends without providing a thorough evaluation of their modeled SMB estimates. Nowadays, comprehensive in situ ablation (Machguth et al. [2016]) and accumulation (Bales et al. [2001, 2009]) data sets are available over Greenland to evaluate modeled SMB in time and space. Such evaluation must be performed and discussed in the manuscript to provide some insight on how well SMB components, i.e. notably precipitation and runoff, are represented in the SnowModel/HydroFlow model. Such evaluation is now systematically performed in Greenland mass balance publications, i.e. Fettweis et al. (2017), Noël et al. (2016), Niwano et al. (2017) or Langen et al. (2017).

2. GrIS-integrated SMB components presented in this study, i.e. notably runoff, meltwater retention and SMB (see Table 1), do not generally agree with recent GrIS SMB studies, e.g. Fettweis et al. [2017], van den Broeke et al. [2016], Noël et al. [2017], Mottram et al. [2017] or Vizcaìno et al. (2013), suggesting potential issues in SnowModel/HydroFlow. For instance, Table 1 shows that only 25% of meltwater is retained in snow while other recent studies suggest 45%, e.g. van Angelen et al. (2013), Noël et al. (2017), Steger et al. (2017a). Steger et al. (2017b) performed a similar basin analysis of SMB components (8 sectors) of the GrIS using another state-of-the-art firn model (SNOWPACK); the authors must compare their results to that study and discuss the differences.

3. Figure 2f strikes me as the ablation zones in north Greenland, i.e. notably in northwest and northeast Greenland are by far larger than in other studies (Fettweis et al., 2017; Noël et al., 2016; Mottram et al., 2017; Vizcaìno et al., 2013; Cullather et al., 2014). In addition, Table 1 suggests that these northern basins contribute equally or more runoff than southern basins, e.g. NW and NE contribute 70 Gt/yr and 63 Gt/yr on average (1979-2014). The study also suggests that these northern basins showed a negative SMB on average (1979-2014), meaning that these have been losing mass for more than 30 years. This is not supported by other studies, e.g. Mouginot et al. (2015) and Steger et al. (2017b).

4. The authors state that they use an updated version of the SnowModel/HydroFlow model but they never discuss the relevant changes implemented, or their impact on the modeled data. The authors should at least list and discuss relevant model updates, compare results from their previous and current model version, and explain where and why changes occur. This would highlight the novelty of the presented data set. To me, it is not clear whether this new data set is a general improvement on previous versions.

5. Tables and figures are sometimes very difficult to read and interpret, especially Figure 5 and Table 1. These may potentially confuse the reader with too much information. Suggestions to improve the text, figures and tables are provided below.

**Point Comments**

**L59-61:** In Wilton et al. (2016), Fig. 2 only shows SMB as low as ~100 Gt/yr in years 2010 and 2012, the same applies for runoff. This should be clarified in the manuscript.

**L65-66:** I would reformulate as follows: "(Chen et al. 2017), and up to 43% for the GrIS and peripheral glaciers and ice caps in 2010-2012 (Noël et al. 2017)".

**L70:** I think the authors mean "melt season duration" instead of "surface ablation duration", see Tedesco et al. (2016).

**L100-102:** I think this sentence is somewhat misleading as Enderlin et al. (2014) compiled estimates of solid ice discharge from ~178 marine-terminating glaciers around Greenland. Could the authors reformulate to clarify this?

**L121:** In the abstract and at L134, the authors state that they use ERA-I reanalysis to force their model, while automatic weather stations are mentioned here. Could the authors clarify and further elaborate on how their model was forced? It would also be useful to learn more about how the snowpack was initialized at the beginning of the simulation.

**Section 2.1:** Here, the authors state that they use an updated version of the SnowModel/HydroFlow model but do not discuss the changes implemented in this new version, nor the impact on the modeled results. See also our Substantive Comment 4.

**L169:** Could the authors clarify what they mean by "water-equivalent evolutions"? I assume this is related to meltwater retention/refreezing in the firn pack and runoff production. Please, elaborate.

**L171:** Could the authors explain what they mean by "hypothetical gridded topography and ocean-mask datasets"?

**L184:** Could the authors briefly explain how the 6 or 12-hourly forcing fields were downscaled to 3-hourly data in MicroMet? Replace "and" by "on" before "a 5-km". Could the precipitation underestimation suggested at L338-343 be the result of this downscaling?

**L188:** Could the authors mention the original resolution of the DEM presented in Levinsen et al. (2015)?

**L213-215:** Why do the authors obtain more catchments? Is this a result of the DEM and/or model updates? Please, clarify.

**L216-223:** Here, the authors describe MicroMet and then resume their discussion on HydroFlow. I would therefore suggest moving these sentences to L201 on page 9.

**L221-222:** Could the authors provide a reference that corroborates this assumption on blowing snow?

**L244:** Could the authors provide a reference for the 10 hydrometric monitoring stations?

**L252:** For model evaluation, the authors refer to a paper that is not published yet. As the authors analyze average SMB components and recent trends, I feel that a proper evaluation, as suggested in the Substantial Comment 1, of the modeled SMB data set must be added here. This would provide some insight on the model performance across the GrIS ablation and accumulation zone, i.e. how well the model simulates runoff and precipitation, respectively. Without model evaluation, the discussion on SMB components and recent trends in Section 4 is somewhat insubstantial.

**L308-317:** These lines are somewhat descriptive, additional insight on the model performance could be gained by performing the SMB evaluation against in situ measurements in the accumulation zone using Bales et al. (2001, 2009).

**L338-351:** Here the authors discuss uncertainties and underestimation of precipitation in the ERA-I forcing field based on, e.g. Fettweis et al. (2017). Again, a proper evaluation of SMB in the accumulation zone would be more convincing to evaluate precipitation and quantify a potential bias that could be further corrected. See also Subtantial Comment 1.

**L366-371:** I don't think that discussing "ablation" adds relevant knowledge to the paper. The authors should better discuss and evaluate melt, runoff and refreezing in more detail. I would also replace ablation by runoff fields in Figs. 2, 4 and Table 1. The authors could consider including these information about "ablation" in a Supplementary Material.

**L375:** The authors probably mean "573.7 +- 119.8 Gt yr-1", which is relatively high compared to other estimates. See also the next comment and Substantial Comment 2.

**L389-392:** The ablation zone in north Greenland, especially in northeast and northwest, are relatively large compared to e.g. RCMs estimate from Fettweis et al. (2017, Fig. 6a), Mottram et al. (2017, Fig. 5) and Noël et al. (2016, Fig. 1), or GCMs estimate from Cullather et al. (2014, Fig. 9) or Vizcaìno et al. (2013, Fig. 7), suggesting ablation overestimate in SnowModel/HydroFlow. Could the authors

elaborate on this? Again, it would be very useful to perform a proper SMB evaluation in the ablation zone of the GrIS. This would allow for estimating SMB (runoff) uncertainty and make the following regional SMB (runoff) analysis more robust. See also Substantive Comment 1.

**L404-406:** I'm not sure to understand these lines, could the authors reformulate?

**L407-419:** The authors obtain a refreezing-retention fraction of 25% for the period 1979-2014, which is by far lower than other estimates of ~45%, e.g. Steger et al. (2017a) or Noël et al. (2017). This could likely explain why runoff is so high compared to other studies, e.g. Van den Broeke et al. (2016). The authors should stress this as these inaccuracies may strongly impact the discussion of contemporary runoff production and recent trends discussed in the paper. In addition, the recent study of Steger et al. (2017a) also integrated refreezing (Gt/yr) and fraction (%) from two state-of-the-art firn models (IMAU-FDM and SNOWPACK) over the same 6 GrIS sections discussed here. For all these sections, the refreezing fraction is lower by almost a factor of 2 compared to Steger et al. (2017a). See also Substantive Comment 2.

**L415:** The authors should refer to Steger et al. (2017a) rather than Ettema et al. (2009).

**L438-439:** Here the authors state that their runoff and SMB product is improved, but no comparison with a previous version or with observations has been conducted. Could the authors elaborate on how they draw these conclusions.

**L442-443:** Given the potential underestimated precipitation, and overestimated runoff, the obtained SMB product is unrealistic. Van den Broeke et al. (2016, Fig. 9) show a reconstruction of GrIS mass balance, solid ice discharge and SMB, clearly refuting an average SMB of ~120 Gt/yr for 1979-2014. This is also supported by other studies: e.g. Fettweis et al. (2017, Fig. 8) and Mottram et al. (2017, Fig. 3). In addition, they obtain a negative SMB after 2005, which is again not supported by other studies.

**L459:** It is misleading to say that Wilton et al. (2016) obtained a GrIS SMB of ~100 Gt/yr in the late-2000s, as this is only true for years 2010 and 2012.

**L480 and 482:** The authors probably refer to Figures 5a and 5b.

**L498-500:** I agree that many catchments show this out-of-phase pattern but there are still quite some that don't. Could the authors quantify this, e.g. as a percentage of the number of catchments in southeast Greenland.

**L509-514:** While I agree that some correlation exists between EOF1 and AMO, it is not so clear for NAO (see Fig. 9b, r =0.4 or $r^2$ =0.16). I would suggest some more caution when drawing firm conclusions on these teleconnections, as at e.g. L44-45, L519 or L544-548.

**L538-540:** This sentence is misleading, to my knowledge no other studies show average SMB of ~120 Gt/yr for 1979-2014, nor a negative SMB in the period 2005-2014. I think the authors should reformulate to stress this.

**L545:** I would replace "indicates" by "suggests" as the correlation obtained for AMO and notably NAO are relatively low. I suggest: "This suggests that runoff variations are related to large-scale natural variability of AMO and NAO in Greenland."

**L549:** My main concern on using the data set presented in this study is that the modeled runoff and SMB are by far overestimated and underestimated, respectively, when compared to other studies. It is therefore questionable whether this data set accurately reproduces the contemporary SMB of the GrIS, if it can be used to force ocean models or to quantify mass changes over Greenland.

**Stylistic comments**

**L33:** I would suggest 'resolution' instead of 'increments'. This holds for the whole manuscript.

**L34:** I suggest: 'Compared to previous studies, simulated SMB is low whereas the GrIS surface conditions remain similar.' In addition, the authors should use the present tense here and at L34-40. Using the past tense is confusing as it suggests that the authors discuss previously published model results.

**L49:** Present tense should also be used in the introduction and following sections when referring to the data discussed in this study.

**L71-72:** I would suggest: "[…] because meltwater may be retained or refrozen in the porous […]".

**L89:** "particularly common".

**L92:** "[…] understanding is used to explain […]".

**L104:** Remove the "of" before "catchments".

**L109:** Remove "the" before "link".

**L110:** I would suggest: "This has further implications […]" as the "unaddressed knowledge gap" is already mentioned in the previous sentence.
**L127:** Maybe replace "land area" by "tundra region".
**L140:** Remove "conditions".
**L141:** Remove "(the last decade […])".
**L150:** I would suggest: "spatiotemporal patterns of runoff".
**L155:** I would replace "verification" by "evaluation".
**L161:** I would suggest: "interpolation scheme. Interpolation fields were adjusted […]"
**L172:** Remove "from" before "catchment outlets".
**L173:** Replace "tested" by "evaluated".
**L189:** I would suggest "resolution" instead of "increment".
**L201:** Replace "of" by "with" before "glacier ice".
**L230:** Remove ", which include a part of the GrIS".
**L231:** I would suggest: "[…], feedbacks from a thinning ice […] will not influence the catchment […]".
**L235-236:** Remove ", not by the glacial drainage system." as this is already mentioned in the previous sentence. L236: Maybe "obtained" instead of "gained".
**L240:** "Evaluation" instead of "Verification". I would also suggest this throughout the whole section (L249, 251).
**L253:** "SMB" instead of "surface mass balance".
**L285:** I would suggest: "The latter analysis enables to link changes in, for example, NAO or AMO with GrIS outlet catchments mass loss and runoff."
**L299 and 301:** Remove "balance" before "loss".
**L319:** Refer to Fig. 1b after "six sections".
**L323-324:** I would suggest: "[…] towards the steep slopes of the southern coast of Greenland, generating orographic enhancement […]".
**L341:** "between 642.0-747.0 Gt yr-1".
**L349:** Maybe: "This highlights the importance of accurately representing precipitation for estimating the energy […]".
**L352-354:** I would suggest: "Besides precipitation, melt (including extent, intensity and duration) and ablation are other […] and understanding GrIS SMB. Surface melt can influence albedo, as wet snow absorbs […]".
**L357:** "[…] affect total runoff, but also ice dynamics […]".
**L366:** The authors certainly mean "northern and southwestern sections".
**L374:** "in southwest Greenland" and for clarity add "over the GrIS" after "period".
**L394:** Maybe "Therefore, in that region the snowpack persists longer compared to […]".
**L402:** Maybe "within the range of our previous study (Mernild et al., 2008)".
**L448:** This sentence could be removed.
**L455:** "24.7 Gt decade-1".
**L484:** Maybe "variance" instead of "variation".
**L489-490:** Replace "goes down/up" by "decreases and increases".
**L515:** This sentence could be removed.

**Figures and Tables**

**Figure 2c and e:** The authors should display regions showing surface melt = 0 in white. This would highlight the dry snow zone of Greenland. The authors should also show runoff instead of ablation using a color scale similar to the one used for melt, i.e. runoff = 0 in white.
**Figure 3:** The scale of SMB components is too small, and numbers are difficult to read. I would also suggest showing values ≤ 0 in white.
**Figure 4:** The authors should better show time series of runoff instead of ablation.
**Figure 5:** This figure is rather overwhelming and confusing. It is very difficult to interpret the data or identify any spatial pattern. In addition, the representation of individual catchment in color is somewhat redundant as it is already shown on Fig. 1c. Therefore, I would suggest to display runoff, variance and trends for each catchment using a color scale instead of circles. For trends, a blue-to-red scale, centered on 0, could be used to distinguish negative from positive values.

**Figure 6:** EOF2 and EOF3 are not significant and the associated figures are not discussed in the paper. Therefore, these could be moved to a Supplementary Material. A new Figure 6 could consist of 3 subpanels combining Figs. 6a, 7 and 8a, all referring to EOF1.

**Figure 8b and c:** These figures could be shown in a Supplementary Material as they are not discussed in the main manuscript.

**Figure 9:** The x-axis of the lower Fig. 9a and b should read "years".

**Table 1:** This Table is rather overwhelming and shows too much information. I think that ablation and "E + Su" could be removed as they are not discussed in detail. In addition, the description of refreezing and retention could be included in the figure caption instead of within the Table itself.

**Additional references and DOI**

- Steger et al. (2017a): https://doi.org/10.3389/feart.2017.00003
- Steger et al. (2017b): https://doi.org/10.5194/tc-11-2507-2017
- Langen et al. (2017): https://doi.org/10.3389/feart.2016.00110
- Mottram et al. (2017): 10.14943/lowtemsci.75.105
- Noël et al. (2016): https://doi.org/10.5194/tc-10-2361-2016
- Niwano et al. (2017): https://doi.org/10.5194/tc-2017-115
- Van den Broeke et al. (2016): https://doi.org/10.5194/tc-10-1933-2016
- Vizcaìno et al. (2013): 10.1175/JCLI-D-12-00615.1
- Cullather et al. (2014): 10.1175/JCLI-D-13-00635.1
- Machguth et al. (2016): 10.1017/jog.2016.75
- Bales et al. (2001): 10.1029/2001JD900153
- Bales et al. (2009): 10.1029/2008JD011208
- Mouginot et al. (2015): 10.1126/science.aac7111

---

## Referee Comment (RC2) · Anonymous Referee #2 · 14 Jan 2018

I fully agree with review #1 that this paper can not be accepted in the present state for publication in TC due to the lack of a robust validation of the presented results. In addition, except Section 4.2 which is a bit interesting, the scientific interest of this paper is very poor and clearly not innovative (e.g. Section 4.1 which is just a confirmation of previous studies)! The mean 1979-2014 SMB over GrIS, the 2012 melt record year and the recent increase of runoff (and corresponding refreeze capacity decrease) in link with changes in AMO and NAO have already been shown and discussed in many papers like the ones from Hanna, Fettweis, Noel, Wilson, . . .. Finally, the studied period

is limited to 2014 while ERA-Interim is available until Oct 2017! The results seem to be a bit outdated.

Line 132-133: The interest of this new version of the SnowModel in respect to the results of Mernild and Liston (2012) should be shown and discussed with validation data.

Line 251: What do you mean by "acceptable results" ?

Line 338-343: only a 2D validation with MAR results should be useful here. A comparison at the scale of the ice sheet is not enough ? Why is there a such underestimation in respect to MAR ? In each area? MAR is not the true and a comparison with the Bales et al. (2001, 2009, . . .) ice cores measurements should be useful here.

Line 316-318: SnowMOdel generally agrees with . . . It is not a validation! What do you mean by "generally agrees" ? A validation with the PROMICE SMB data set is absolutely needed.

Fig 2: showing mean SMB without a validation or without a 2D comparison with other models (e.g. MAR or RACMO) is not interesting for me.

Fig 3b: what is the interest of showing the 2 last figures?

Fig5: these figures are not readable.

---

## Author Comment (AC1) · 10 Apr 2018

To the Editor of The Cryosphere April 10, 2018

Dear Editor,

First of all, thank you for your mail by 4 February 2017 and for the valuable and thorough comments from the reviewers. We apologize for the delay of our reply.

Included below are our comments for the paper: "Reconstruction of the Greenland Ice

[Figure]

Sheet surface mass balance and the spatiotemporal distribution of freshwater runoff from Greenland to surrounding seas" by S. H. Mernild, G. E. Liston, A. P. Beckerman, and J. C. Yde based on the comments from the reviewers.

On the next pages, the reviewers' comments are shown with blue color and our responses to the reviewers' comments are shown with black color.

After the submission of this manuscript to TC (autumn 2017), a study by Mernild et al. was published (https://www.tandfonline.com/doi/full/10. 1080/15230430.2017.1415856). This study included a detailed evaluation of SnowModel/HydroFlow simulated GrIS air temperature, SMB, ELA, and catchment outlet river runoff for the Kangerlussuaq sub-catchment in west Greenland. The same model setup, forcing and DEM were used in both studies, but on different domains. The reviewers requested a SnowModel GrIS evaluation, and this is now available (see more below).

Thank you for your help. If we can be of any further assistance, please feel free to contact me.

Best regards Sebastian H. Mernild and co-authors

——

Anonymous Referee #1

Review of: "Reconstruction of the Greenland Ice Sheet surface mass balance and spatiotemporal distribution of freshwater runoff from Greenland to surrounding seas", by S. H. Mernild et al., submitted to The Cryosphere.

General comments Using the updated version of SnowModel/HydroFlow, the authors simulate the surface mass balance (SMB) and components, i.e. runoff, melt and retention, of the Greenland ice sheet (GrIS) at 5 km resolution for the period 1979-2014. Precipitation is downscaled from ERA-Interim re-analysis for the same period. The model includes a snow module accounting for meltwater retention in snow, an energy

balance scheme and a meltwater runoff routing module. This routing module allows to quantify the runoff contribution of 6 GrIS sectors, further refined to over 3,000 individual catchments. The authors first discuss the modeled contemporary (1979-2014) GrIS SMB and recent trends (2005-2014) in these 6 sectors. The analysis is further extended to the numerous individual catchments to show that about 80% of these have experienced increasing meltwater runoff since 1979. Then, the authors correlate this recent runoff increase with the natural variability of the Atlantic Multidecadal Oscillation (AMO) and North Atlantic Oscillation (NAO).

The manuscript is overall well written, but the results presented seem inaccurate and raise questions and concerns on the model ability to reproduce the contemporary GrIS SMB, potentially altering the conclusions drawn in this paper. These concerns are summarized in the Substantive Comments. In brief, I suspect that the current version of SnowModel/HydroFlow has several issues resulting in inaccurate SMB estimates over the GrIS. Compared to other studies, runoff is significantly overestimated likely due to inaccurate representation of meltwater retention in firn. This study suggests that only 25% of melt is retained in the firn pack while most recent efforts demonstrated that it is closer to ∼45%. Such a retention underestimation has severe implications on runoff calculation and SMB. For instance, the authors present negative modeled GrIS-integrated SMB for the period 2005-2014 as opposed to recently published GrIS mass balance and GRACE studies. This makes these results potentially unreliable and the conclusions drawn inconsistent. Authors: We are grateful to the reviewer for pointing out the overestimation in runoff caused by underestimation of meltwater retention in the firn pack. The model simulations were tested and evaluated by Mernild et al. (2018) based on long-term observations from the GrIS. Due to the GrIS surface water balance components (using the hydrological method, the continuity equation): P – (Su + E) – R + ∆S = 0 ± ðÍIJĆ, a change in one parameter effects ∆S, also referred to as SMB. The evaluation against runoff observations from the GrIS have caused adjustments to SMB and retention/refreezing. Therefore, the text (Results and discussion section), Figures and Tables have been updated since the last version of this manuscript. For

the GrIS, we find that the 35-year mean refreezing and retention was estimated to be 49 % (269.9 ± 77.4 Gt yr-1), and it was 45 % (318.0 ± 62.8 Gt yr-1) for the period 2005–2014. The Substantive Comments have been addressed below.

For these reasons, I judge that the manuscript cannot be published in the current form and needs major revisions, unless the authors prove that their estimates of precipitation, runoff, melt, refreezing and SMB are reasonably accurate. To achieve this, the authors must perform a thorough model evaluation against in situ SMB measurements and compare their results to remote sensing mass change records, and mass balance estimates compiled in previously published studies. This would highlight potential issues in their snow model and provide some insight on how to solve them. If these evaluations/comparisons and validation of the model results can be successfully achieved, and if the suggested corrections listed hereunder are applied, I would be happy to reassess this manuscript.

Substantive Comments 1 Throughout the manuscript, the authors discuss changes in SMB components and recent trends without providing a thorough evaluation of their modeled SMB estimates. Nowadays, comprehensive in situ ablation (Machguth et al. [2016]) and accumulation (Bales et al. [2001, 2009]) data sets are available over Greenland to evaluate modeled SMB in time and space. Such evaluation must be performed and discussed in the manuscript to provide some insight on how well SMB components, i.e. notably precipitation and runoff, are represented in the SnowModel/ HydroFlow model. Such evaluation is now systematically performed in Greenland mass balance publications, i.e. Fettweis et al. (2017), Noel et al. (2016), Niwano et al. (2017) or Langen et al. (2017). Authors: We thank the reviewer for bringing this to our attention. SnowModel/HydroFlow evaluations have been conducted against independent long-term observations from the Kangerlussuaq GrIS catchment, e.g. against air temperature observed at AWS located on the GrIS (at the K-transect for AWS: S5, S6, and S9 to illustrate the elevation variability in MAAT; van den Broeke et al. 2008a, 2008b; van de Wal et al. 2005), GrIS SMB observations to illustrate the elevation variability

(K-transect for AWS: S4–S9 and SHR; van den Broeke et al. 2008a, 2008b; van de Wal et al. 2005), GrIS ELA estimations (van de Wal et al. (2012); van As et al. (2017), and observed GrIS catchment outlet river runoff (Hasholt et al. (2013); van As et al. (2017)). The Kangerlussuaq GrIS catchment is the best observed catchment in Greenland – and therefore, it was used as a test site. These evaluations have been published in Mernild et al. (2018, doi.org/10.1080/15230430.2017.1415856). The Mernild et al. (2018) study uses the exact same model settings, forcing, and DEM as this study. The Mernild et al. (2018) study can be seen as a Part 1, where this study can be seen as Part 2 (similar Part 1 and 2 studies were seen in Liston and Mernild (2012, JCLI) and Mernild and Liston (2012, JCLI), where Part 1 was an evaluation of the model and Part 2 an application of the model). In this TC manuscript, we follow the advice from the reviewer by evaluation SnowModel/HydroFlow against GrIS observations from Mernild et al. 2018. We used observed GrIS independent values for validation – we did not use other model simulations for evaluation. However, in the Result and Discussion section we discussed our simulated results of SMB and retention/refreezing against outputs from other model studies, see Section 4.1.

2 GrIS-integrated SMB components presented in this study, i.e. notably runoff, meltwater retention and SMB (see Table 1), do not generally agree with recent GrIS SMB studies, e.g. Fettweis et al. [2017], van den Broeke et al. [2016], Noel et al. [2017], Mottram et al. [2017] or Vizcaino et al. (2013), suggesting potential issues in SnowModel/HydroFlow. For instance, Table 1 shows that only 25% of meltwater is retained in snow while other recent studies suggest 45%, e.g. van Angelen et al. (2013), Noel et al. (2017), Steger et al. (2017a). Steger et al. (2017b) performed a similar basin analysis of SMB components (8 sectors) of the GrIS using another state-of-the-art firn model (SNOWPACK); the authors must compare their results to that study and discuss the differences. Authors: We agree with the reviewer. Therefore, simulated runoff was evaluated against observations from the Kangerlussuaq catchment and adjusted. In the study by Mernild et al. (2018) simulated runoff was on average overestimated by 31 % (2007-2014) compared to observations. This was likely because of missing multiyear firn processes, such as nonlinear meltwater retention, percolation blocked by ice layers, and refreezing in SnowModel, more specifically in the submodel SnowPack-ML. Therefore, based on the findings from the Kangerlussuaq catchment, we have now adjusted runoff due to the missing firn routines, and subsequently retention/refreezing and SMB were changed. After the GrIS runoff, retention/refreezing, and SMB adjustments, the 35-year (1979-2014) runoff, retention/refreezing and SMB values seems in line with earlier studies, although SMB was in the low end compared to other model SMB simulations (see Section 4.1).

3 Figure 2f strikes me as the ablation zones in north Greenland, i.e. notably in northwest and northeast Greenland are by far larger than in other studies (Fettweis et al., 2017; Noel et al., 2016; Mottram et al., 2017; Vizcaino et al., 2013; Cullather et al., 2014). In addition, Table 1 suggests that these northern basins contribute equally or more runoff than southern basins, e.g. NW and NE contribute 70 Gt/yr and 63 Gt/yr on average (1979-2014). The study also suggests that these northern basins showed a negative SMB on average (1979-2014), meaning that these have been losing mass for more than 30 years. This is not supported by other studies, e.g. Mouginot et al. (2015) and Steger et al. (2017b). Authors: After adjustments and recalculations of GrIS runoff, retention/refreezing and SMB (Figures and Tables were updated), our values were compared to earlier studies and discussed in Section 4.1. Mean refreezing and retention was estimated to be 49 % (1979-2014), and 45 % (2005-2014), and not around 25% as in our earlier version of this manuscript. These values (49 and 45 %) are more in line with previous studies. A comparison and discussion of SnowModel simulated GrIS SMB values were further done against other SMB studies (section 4.1).

4 The authors state that they use an updated version of the SnowModel/HydroFlow model but they never discuss the relevant changes implemented, or their impact on the modeled data. The authors should at least list and discuss relevant model updates, compare results from their previous and current model version, and explain where and why changes occur. This would highlight the novelty of the presented data set. To me,

it is not clear whether this new data set is a general improvement on previous versions. Authors: We agree with the reviewer that the implications of using the improved model should be better clarified. The text was rewritten to make this clearer. The novelty of this work can be summarized by the following bullet points: 1) never before we have simulated GrIS SMB for present day conditions using the ERA-Interim (ERA-I) reanalysis atmospheric forcing (in earlier simulations, only observed meteorological data from AWS were used); 2) we implimented retention/refreezing conditions in more detail than in earlier SnowModel GrIS studies; 3) we simulated spatial runoff in more detailed than earlier; and 4) we improved the understanding of the spatiotemporal distribution of freshwater river runoff from Greenland to the ocean using the statistical EOF method. The spatiotemporal distribution of runoff is important because the GrIS and Greenland plays an essential role in the Arctic hydrological cycle and for the individual catchment budgets, where river runoff is the hydrological link between snowmelt and ice melt and hydrographic and circulation conditions in fjords and the adjacent seas.

5 Tables and figures are sometimes very difficult to read and interpret, especially Figure 5 and Table 1. These may potentially confuse the reader with too much information. Suggestions to improve the text, figures and tables are provided below. Authors: Figure 5 and Table 1 were updated and changed. We hope that this has improved the readability. Please see further points below.

Point Comments L59-61: In Wilton et al. (2016), Fig. 2 only shows SMB as low as ∼100 Gt/yr in years 2010 and 2012, the same applies for runoff. This should be clarified in the manuscript. Authors: The text was rewritten following the advice from the reviewer.

L65-66: I would reformulate as follows: "(Chen et al. 2017), and up to 43% for the GrIS and peripheral glaciers and ice caps in 2010-2012 (Noel et al. 2017)". Authors: The text was rewritten following the advice from the reviewer.

L70: I think the authors mean "melt season duration" instead of "surface ablation duration", see Tedesco et al. (2016). Authors: Yes, thanks. This was fixed.

L100-102: I think this sentence is somewhat misleading as Enderlin et al. (2014) compiled estimates of solid ice discharge from ∼178 marine-terminating glaciers around Greenland. Could the authors reformulate to clarify this? Authors: The text was rewritten.

L121: In the abstract and at L134, the authors state that they use ERA-I reanalysis to force their model, while automatic weather stations are mentioned here. Could the authors clarify and further elaborate on how their model was forced? It would also be useful to learn more about how the snowpack was initialized at the beginning of the simulation. Authors: Here, we talk about how the study by Mernild and Liston (2012) was forced with data from AWS. The present study was forced with ERA-I. The text was rewritten to clarify that Mernild and Liston (2012) was forced with data from AWS. The snowpack was initialized in September 1 each year.

Section 2.1: Here, the authors state that they use an updated version of the Snow-Model/ HydroFlow model but do not discuss the changes implemented in this new version, nor the impact on the modeled results. See also our Substantive Comment 4. Authors: The text was rewritten to clarify this.

L169: Could the authors clarify what they mean by "water-equivalent evolutions"? I assume this is related to meltwater retention/refreezing in the firn pack and runoff production. Please, elaborate. Authors: The text is rewritten to make this clearer.

L171: Could the authors explain what they mean by "hypothetical gridded topography and oceanmask datasets"? Authors: The word 'hypothetical' was erased. The text is rewritten to highlight that the runoff configuration was based on gridded topography.

L184: Could the authors briefly explain how the 6 or 12-hourly forcing fields were downscaled to 3-hourly data in MicroMet? Replace "and" by "on" before "a 5-km". Could the precipitation underestimation suggested at L338-343 be the result of this
downscaling? Authors: The downscaling to 3-hours is now explained, and the word 'and' was changed to 'on'. No, the downscaling to 3-hours has nothing to do with the underestimating mentioned at L338-343.

L188: Could the authors mention the original resolution of the DEM presented in Levinsen et al. (2015)? Authors: The original resolution was 4 kmˆ2; 2 × 2 km. This information was added to the text.

L213-215: Why do the authors obtain more catchments? Is this a result of the DEM and/or model updates? Please, clarify. Authors: Yes, this is a result of the new DEM.

L216-223: Here, the authors describe MicroMet and then resume their discussion on HydroFlow. I would therefore suggest moving these sentences to L201 on page 9. Authors: Is done.

L221-222: Could the authors provide a reference that corroborates this assumption on blowing snow? Authors: Reference is added.

L244: Could the authors provide a reference for the 10 hydrometric monitoring stations? Authors: Reference is added.

L252: For model evaluation, the authors refer to a paper that is not published yet. As the authors analyze average SMB components and recent trends, I feel that a proper evaluation, as suggested in the Substantial Comment 1, of the modeled SMB data set must be added here. This would provide some insight on the model performance across the GrIS ablation and accumulation zone, i.e. how well the model simulates runoff and precipitation, respectively. Without model evaluation, the discussion on SMB components and recent trends in Section 4 is somewhat insubstantial. Authors: The mentioned paper is now published: Mernild et al. (2018), doi.org/10.1080/15230430.2017.1415856. In this paper, an evaluation is done for the only GrIS catchment where long-term observed AWS data, GrIS SMB data, and catchment runoff data are available. The model evaluation has been mentioned in Section

2.3.

L308-317: These lines are somewhat descriptive, additional insight on the model performance could be gained by performing the SMB evaluation against in situ measurements in the accumulation zone using Bales et al. (2001, 2009). Authors: Sure, we did additional SMB comparisons using e.g., Bales et al. (2001, 2009), see Section 4.1.

L338-351: Here the authors discuss uncertainties and underestimation of precipitation in the ERA-I forcing field based on, e.g. Fettweis et al. (2017). Again, a proper evaluation of SMB in the accumulation zone would be more convincing to evaluate precipitation and quantify a potential bias that could be further corrected. See also Subtantial Comment 1. Authors: An evaluation of SMB was shown and discussed in the study by Mernild et al. (2018) using observed SMB data from AWS from the K-transect, covering the period from 1990-2014. This evaluation was done for the K-transect stations (S4, S5, SHR, S6, S7, S8, S9, and S10) located in elevation from c. 340 m a.s.l. to c. 1,850 m a.s.l., where AWS S9 likely are located in the accumulation zone, since ELA is within the elevation range of 1,610-1,800 m a.s.l. (van de Wal et al. 2012; van As et al. 2017). Further, an evaluation of SnowModel simulated ELA was done (the location of ELA is based on both accumulation and ablation processes), showing 35-year mean ELA simulated values (1,760 $\pm$ 260 m a.s.l.) to be within the range of ELA values from other studies 1,610-1,800 m a.s.l. (van de Wal et al. 2012; van As et al. 2017) within the Kangerlussuaq catchment, SW Greenland.

L366-371: I don't think that discussing "ablation" adds relevant knowledge to the paper. The authors should better discuss and evaluate melt, runoff and refreezing in more detail. I would also replace ablation by runoff fields in Figs. 2, 4 and Table 1. The authors could consider including these information about "ablation" in a Supplementary Material. Authors: Ablation as well as accumulation are important processes to understand the GrIS SMB conditions. Therefore, we keep the discussion of ablation in the paper. We have added runoff time series to Figure 4 and Table 1, as runoff is an important part of the ablation, but also to show all the water balance components together (Equation

1, Section 2.4).

L375: The authors probably mean "573.7 +- 119.8 Gt yr-1", which is relatively high compared to other estimates. See also the next comment and Substantial Comment 2. Authors: The reviewer is correct. The manuscript has been updated after Mernild et al. (2018) was published, after the evaluation and adjustment. Further, see comments after the Substantial Comment #2.

L389-392: The ablation zone in north Greenland, especially in northeast and north-west, are relatively large compared to e.g. RCMs estimate from Fettweis et al. (2017, Fig. 6a), Mottram et al. (2017, Fig. 5) and Noel et al. (2016, Fig. 1), or GCMs estimate from Cullather et al. (2014, Fig. 9) or Vizcaino et al. (2013, Fig. 7), suggesting ablation overestimate in SnowModel/ HydroFlow. Could the authors elaborate on this? Again, it would be very useful to perform a proper SMB evaluation in the ablation zone of the GrIS. This would allow for estimating SMB (runoff) uncertainty and make the following regional SMB (runoff) analysis more robust. See also Substantive Comment 1. Authors: Absolutely, and that is why SMB was compared to observed SMB values from the K-transect, also covering elevations ranging into the ablation zone, below around 1,600 m.asl. (AWS S4-S9 and SHR). In the Result and Discussion section, we compare SnowModel simulated SMB against other studies. Further, see comments after the Substantial Comment #1.

L404-406: I'm not sure to understand these lines, could the authors reformulate? Authors: These lines were erased to avoid any confusion.

L407-419: The authors obtain a refreezing-retention fraction of 25% for the period 1979-2014, which is by far lower than other estimates of ∼45%, e.g. Steger et al. (2017a) or Noel et al. (2017). This could likely explain why runoff is so high compared to other studies, e.g. Van den Broeke et al. (2016). The authors should stress this as these inaccuracies may strongly impact the discussion of contemporary runoff production and recent trends discussed in the paper. In addition, the recent study of Steger et
al. (2017a) also integrated refreezing (Gt/yr) and fraction (%) from two state-of-the-art firn models (IMAU-FDM and SNOWPACK) over the same 6 GrIS sections discussed here. For all these sections, the refreezing fraction is lower by almost a factor of 2 compared to Steger et al. (2017a). See also Substantive Comment 2. Authors: Sure, we are aware about this, but after the evaluation and adjustment in Mernild et al. (2018) we end up with a refreezing-retention fraction of 49% (1979-2014) and 45% (2005-2014). These values are more in line with other studies. This is discussed in the text.

L415: The authors should refer to Steger et al. (2017a) rather than Ettema et al. (2009). Authors: Sure, is fixed.

L438-439: Here the authors state that their runoff and SMB product is improved, but no comparison with a previous version or with observations has been conducted. Could the authors elaborate on how they draw these conclusions. Authors: These lines were erased. Please see our response to substantive comment 4.

L442-443: Given the potential underestimated precipitation, and overestimated runoff, the obtained SMB product is unrealistic. Van den Broeke et al. (2016, Fig. 9) show a reconstruction of GrIS mass balance, solid ice discharge and SMB, clearly refuting an average SMB of ∼120 Gt/yr for 1979-2014. This is also supported by other studies: e.g. Fettweis et al. (2017, Fig. 8) and Mottram et al. (2017, Fig. 3). In addition, they obtain a negative SMB after 2005, which is again not supported by other studies. Authors: This sentence was reformulated due to the updated (evaluated/adjusted) Snow-Model/HydroFlow simulations and calculations, based on the Mernild et al. (2018) study.

L459: It is misleading to say that Wilton et al. (2016) obtained a GrIS SMB of ∼100 Gt/yr in the late-2000s, as this is only true for years 2010 and 2012. Authors: Thanks for making this mistake clear to us. The text is corrected and the years 2010 and 2012 are used instead.

L480 and 482: The authors probably refer to Figures 5a and 5b. Authors: Yes, the

reviewer is right.

L498-500: I agree that many catchments show this out-of-phase pattern but there are still quite some that don't. Could the authors quantify this, e.g. as a percentage of the number of catchments in southeast Greenland. Authors: This is now done, and it is around 20 %.

L509-514: While I agree that some correlation exists between EOF1 and AMO, it is not so clear for NAO (see Fig. 9b, r =0.4 or r2 =0.16). I would suggest some more caution when drawing firm conclusions on these teleconnections, as at e.g. L44-45, L519 or L544-548. Authors: We erased the word 'strong' related to the correlation, to be more caution when drawing conclusions (related to L509-514). Also, other places where mentioned by the reviewer (L44-45, L519 or L544-548) the text was carefully looked through to be more caution when drawing conclusions on these teleconnections.

L538-540: This sentence is misleading, to my knowledge no other studies show average SMB of ∼120 Gt/yr for 1979-2014, nor a negative SMB in the period 2005-2014. I think the authors should reformulate to stress this. Authors: This sentence was reformulated due to the updated (evaluated/adjusted) SnowModel/HydroFlow simulations and calculations, based on the Mernild et al. (2018) study.

L545: I would replace "indicates" by "suggests" as the correlation obtained for AMO and notably NAO are relatively low. I suggest: "This suggests that runoff variations are related to large-scale natural variability of AMO and NAO in Greenland." Authors: Is fixed.

L549: My main concern on using the data set presented in this study is that the modeled runoff and SMB are by far overestimated and underestimated, respectively, when compared to other studies. It is therefore questionable whether this data set accurately reproduces the contemporary SMB of the GrIS, if it can be used to force ocean models or to quantify mass changes over Greenland. Authors: We see the concern of the reviewer. In the catchment-scale study by Mernild et al. (2018,

doi.org/10.1080/15230430.2017.1415856) (see also other places in this reply letter) GrIS surface conditions and Greenland runoff conditions were tested against independent observations, from where simulations were evaluated and subsequent adjusted. Simulated runoff was adjusted against observations and subsequent calculations were redone.

Stylistic comments L33: I would suggest 'resolution' instead of 'increments'. This holds for the whole manuscript. Authors: Is fixed.

L34: I suggest: 'Compared to previous studies, simulated SMB is low whereas the GrIS surface conditions remain similar.' In addition, the authors should use the present tense here and at L34-40. Using the past tense is confusing as it suggests that the authors discuss previously published model results. Authors: The first part of the comment was erased from the abstract. Present tense was used in the abstract.

L49: Present tense should also be used in the introduction and following sections when referring to the data discussed in this study. Authors: This is an interesting question raised by the reviewer. Basically, the rule of thumb is when refereeing to studies which had happened (already been published) the text should be written in past tense. If one is describing a figure or a table, then the text should be in present tense.

L71-72: I would suggest: "[. . .] because meltwater may be retained or refrozen in the porous [. . .]". Authors: Is fixed.

L89: "particularly common". Authors: Is fixed.

L92: "[. . .] understanding is used to explain [. . .]". Authors: Is fixed.

L104: Remove the "of" before "catchments". Authors: Is fixed.

L109: Remove "the" before "link". Authors: Is fixed.

L110: I would suggest: "This has further implications [. . .]" as the "unaddressed knowledge gap" is already mentioned in the previous sentence. Authors: Is fixed partly.

L127: Maybe replace "land area" by "tundra region". Authors: Is fixed.

L140: Remove "conditions". Authors: Is fixed.

L141: Remove "(the last decade [. . .])". Authors: Is fixed.

L150: I would suggest: "spatiotemporal patterns of runoff". Authors: Is fixed.

L155: I would replace "verification" by "evaluation". Authors: Is fixed.

L161: I would suggest: "interpolation scheme. Interpolation fields were adjusted [. . .]" Authors: Is fixed.

L172: Remove "from" before "catchment outlets". Authors: Is fixed.

L173: Replace "tested" by "evaluated". Authors: Is fixed.

L189: I would suggest "resolution" instead of "increment". Authors: Is fixed.

L201: Replace "of" by "with" before "glacier ice". Authors: Is fixed.

L230: Remove ", which include a part of the GrIS". Authors: Is fixed.

L231: I would suggest: "[. . .], feedbacks from a thinning ice [. . .] will not influence the catchment [. . .]". Authors: Is fixed.

L235-236: Remove ", not by the glacial drainage system." as this is already mentioned in the previous sentence. L236: Maybe "obtained" instead of "gained". Authors: Is fixed.

L240: "Evaluation" instead of "Verification". I would also suggest this throughout the whole section (L249, 251). Authors: Is fixed.

L253: "SMB" instead of "surface mass balance". Authors: Is fixed.

L285: I would suggest: "The latter analysis enables to link changes in, for example, NAO or AMO with GrIS outlet catchments mass loss and runoff." Authors: Is fixed.

[Figure]

L299 and 301: Remove "balance" before "loss". Authors: Is fixed.

L319: Refer to Fig. 1b after "six sections". Authors: Is fixed.

L323-324: I would suggest: "[. . .] towards the steep slopes of the southern coast of Greenland, generating orographic enhancement [. . .]". Authors: Is fixed.

L341: "between 642.0-747.0 Gt yr-1". Authors: Is fixed.

L349: Maybe: "This highlights the importance of accurately representing precipitation for estimating the energy [. . .]". Authors: Is fixed.

L352-354: I would suggest: "Besides precipitation, melt (including extent, intensity and duration) and ablation are other [. . .] and understanding GrIS SMB. Surface melt can influence albedo, as wet snow absorbs [. . .]". Authors: Is fixed.

L357: "[. . .] affect total runoff, but also ice dynamics [. . .]". Authors: Is fixed.

L366: The authors certainly mean "northern and southwestern sections". Authors: Is fixed.

L374: "in southwest Greenland" and for clarity add "over the GrIS" after "period". Authors: Is fixed.

L394: Maybe "Therefore, in that region the snowpack persists longer compared to [. . .]". Authors: Is fixed.

L402: Maybe "within the range of our previous study (Mernild et al., 2008)". Authors: Is fixed.

L448: This sentence could be removed. Authors: Is erased.

L455: "24.7 Gt decade-1". Authors: Is fixed.

L484: Maybe "variance" instead of "variation". Authors: Is fixed.

L489-490: Replace "goes down/up" by "decreases and increases". Authors: Is fixed.

L515: This sentence could be removed. Authors: Is fixed.

Figures and Tables Figure 2c and e: The authors should display regions showing surface melt = 0 in white. This would highlight the dry snow zone of Greenland. The authors should also show runoff instead of ablation using a color scale similar to the one used for melt, i.e. runoff = 0 in white. Authors: Figure 2c: Since this is 35-year mean spatial surface melt the grid values will never be equal to zero if just melt occurred a single day. That is the reason why we used the purple color for mean annual melt values below 0.0625 m w.e. Figure 2e: we don't see the issues here – it could be ablation (R+Su+E) and/or runoff (R) alone displayed. Since Su+E is relatively low, ablation and runoff would be rather insignificant difference. However, in Figure 4, the runoff time series (shown in red color) are shown together with retention/refreezing time series.

Figure 3: The scale of SMB components is too small, and numbers are difficult to read. I would also suggest showing values $\leq 0$ in white. Authors: The numbers were made bigger in size. It seems not proper to show refreezing/retention values $\leq 0$ as white color, since the ocean around Greenland already is white. It can provide misunderstandings.

Figure 4: The authors should better show time series of runoff instead of ablation. Authors: Now runoff time series are shown together with ablation time series.

Figure 5: This figure is rather overwhelming and confusing. It is very difficult to interpret the data or identify any spatial pattern. In addition, the representation of individual catchment in color is somewhat redundant as it is already shown on Fig. 1c. Therefore, I would suggest to display runoff, variance and trends for each catchment using a color scale instead of circles. For trends, a blue-to-red scale, centered on 0, could be used to distinguish negative from positive values. Authors: The representation of individual catchments in colors are erased due to avoid redundancy. For each catchment, a circle is shown where the size depends on the mean runoff volume. The suggestion

by the reviewer of making runoff, variance, and trends following a color scale bar was tested, but it seems even more confusing (it is very difficult to differentiating the different colors) and this gives even less overview of the differences between catchments (when plotting it is even more difficult to get an overview of the runoff variability from 3000+ catchments). After careful considerations, we have therefore decided to keep the runoff volume illustrated as circles. Here, positive trends are shown in red color, and negative trends in blue. The negative trends are placed on top of the positive trends to better highlight the differences between increasing and decreasing trends.

Figure 6: EOF2 and EOF3 are not significant and the associated figures are not discussed in the paper. Therefore, these could be moved to a Supplementary Material. A new Figure 6 could consist of 3 subpanels combining Figs. 6a, 7 and 8a, all referring to EOF1. Authors: Sure, a new Figure 6 was established (combining the 'old' figures 6a, 7, and 8a).

Figure 8b and c: These figures could be shown in a Supplementary Material as they are not discussed in the main manuscript. Authors: All figures related to EOF2 and EOF3 was placed in a supplementary material.

Figure 9: The x-axis of the lower Fig. 9a and b should read "years". Authors: Is fixed.

Table 1: This Table is rather overwhelming and shows too much information. I think that ablation and "E + Su" could be removed as they are not discussed in detail. In addition, the description of refreezing and retention could be included in the figure caption instead of within the Table itself. Authors: To understand the water balance components, we think that it is important to still keep 'E+Su' in the Table. Otherwise, a part of the water balance is not shown. We followed the advice and removed the description of refreezing and retention. The description is now in the caption.

Additional references and DOI  c Steger et al. (2017a): https://doi.org/10.3389/feart.2017.00003  c Steger et al. (2017b): https://doi.org/10.5194/tc-11-2507-2017  c Langen et al. (2017):

https://doi.org/10.3389/feart.2016.00110 • Mottram et al. (2017): 10.14943/lowtem-sci.75.105 • Noel et al. (2016): https://doi.org/10.5194/tc-10-2361-2016 • Niwano et al. (2017): https://doi.org/10.5194/tc-2017-115 • Van den Broeke et al. (2016): https://doi.org/10.5194/tc-10-1933-2016 • Vizcaino et al. (2013): 10.1175/JCLI-D-12-00615.1 • Cullather et al. (2014): 10.1175/JCLI-D-13-00635.1 • Machguth et al. (2016): 10.1017/jog.2016.75 • Bales et al. (2001): 10.1029/2001JD900153 • Bales et al. (2009): 10.1029/2008JD011208 • Mouginot et al. (2015): 10.1126/science.aac7111 Authors: Some of these published studies are referred to in the text, where we found it relevant.

(a)

(b)

N

NW    NE

W

SE

SW

(c)

**Fig. 1.** Figure 1

[Figure]

**Fig. 2.** Figure 2

[Figure]

**Fig. 3.** Figure 3

[Figure]

**Fig. 4.** Figure 4

(a)

Mean

Variance

Trend

N

NE

NW

W

SE

SW

7.5

-2

-4

(b)

40

80

-2

-4

**Fig. 5.** Figure 5

[Figure]

(a)

[Figure]

(b)

[Figure]

(c)

[Figure]

**Fig. 6.** Figure 6

[Figure]

[Figure]

[Figure]

**Fig. 7.** Figure 7

[Figure]

**Fig. 8.** Figure S1

[Figure]

**Fig. 9.** Figure S2

---

## Author Comment (AC2) · 10 Apr 2018

I fully agree with review #1 that this paper can not be accepted in the present state for publication in TC due to the lack of

a robust validation of the presented results. In addition, except Section 4.2 which is a bit interesting, the scientific interest of this paper is very poor and clearly not innovative (e.g. Section 4.1 which is just a confirmation of previous studies)! The mean 1979-2014 SMB over GrIS, the 2012 melt record year and the recent increase of runoff (and corresponding refreeze capacity decrease) in link with changes in AMO and NAO have already been shown and discussed in many papers like the ones from Hanna, Fettweis, Noel, Wilson, : : :. Finally, the studied period is limited to 2014 while ERA-Interim is available until Oct 2017! The results seem to be a bit outdated. Authors: In the papers mentioned by the reviewer the spatiotemporal distribution of individual catchment runoff from Greenland have not been studied, and linked to AMO and NAO. This is main scientific novelty of this study. SnowModel/HydroFlow is unique for simulation of the spatiotemporal distribution from each individual catchment, which no other model can do at present. This is important if we want to understand and link the terrestrial runoff freshwater from Greenland to ocean dynamic models, suspended sediment fluxes or biogeochemical/nutrient fluxes. The EOF analyzes and it potential freshwater runoff link to other scientific communities is of high interest and innovative, for better understanding how the terrestrial environment links to the marine environment. Here, we present 35-years (1979-2014) for spatiotemporal distribution of freshwater runoff from Greenland (understanding the distribution from a statistical perspective – an EOF perspective), and in that perspective adding three more years is not significant.

Line 132-133: The interest of this new version of the SnowModel in respect to the results of Mernild and Liston (2012) should be shown and discussed with validation data. Authors: The text was rewritten and parts were erased. Please see our responses to the comments by Reviewer #1.

Line 251: What do you mean by "acceptable results" ? Authors: This is now explained.

Line 338-343: only a 2D validation with MAR results should be useful here. A comparison at the scale of the ice sheet is not enough ? Why is there a such underestimation in respect to MAR ? In each area? MAR is not the true and a comparison

with the Bales et al. (2001, 2009, : : :) ice cores measurements should be useful here. Authors: An evaluation against other model outputs is not the most optimal way of doing evaluations. We evaluated our simulations against independent GrIS observations (see our answers to Reviewer #1). The simulations were evaluated in detail against observations from the GrIS Kangerlussuaq catchment, from where we have the most detailed long-term observations on air temperature, SMB, location of ELA, and catchment outlet river runoff available. This evaluation has been included in Section 2.3, and can further be found in the recent published paper by Mernild et al. (2018), doi.org/10.1080/15230430.2017.1415856. For further see above.

Line 316-318: SnowMOdel generally agrees with : : : It is not a validation! What do you mean by "generally agrees" ? A validation with the PROMICE SMB data set is absolutely needed. Authors: We agree that validation is important, which is already highlighted in the text (See Section 2.3), and why we several times in previous GrIS studies have verified SnowModel GrIS simulations against independent observations, to highlight the (in)accuracy of our simulations. The present SnowModel ERA-I GrIS simulations (using the same DEM) was validated against observations from the Kangerlussuaq catchment, SW Greenland, against observations from AWS and SMB from the K-transect and catchment outlet runoff (Mernild et al. 2018).

Further, below is a list of the SnowModel GrIS studies, where SnowModel routines have been validated over time against independent observations:

Parameters used for validation Difference between simulations and observations Time period Reference Meteorological data from AWS located on GrIS (from GC-Net): - Air temperature - Wind speed - Relatively humidity - precipitation

$0.2°C$ 0.2 m s-1 0.1 % 1.0 mm w.e. 1995-2005 Mernild et al. 2008

MAAT (from K-Transect) $0.2–0.5°C$ (Forcing and DEM are similar to this TCD study in review) 1979-2014 Mernild et al. 2018 End-of-Seacon satellite-derived surface melt extent Average 4 % and maximum distance of 160 km between modeled and satellitederived melt and non-melt boundaries. 1995-2005 Mernild et al. 2008

End-of-season satellite-derived surface melt extent

Mean annual satellite-derived surface melt extent Average 40 ± 35 km and maximum
âĄŞ160 km

0.4 × 105 km2 2010

1979-2010 Mernild et al. 2011 Melt-index Modeled results are consistent with observations 1995-2005 Mernild et al. 2008 Location of ELA Average 35 m a.s.l. and maximum 425 m a.s.l. 1995-2005 Mernild et al. 2008 Average meteorological data (the explained variance): - air temperature - wind speed - precipitation - relatively humidity

SWE depth

End-of-season satellite-derived surface melt extent

84-87% 49-55 % 49-69 % 48-63 %

1 %

Average 7.8 ± 5.1 and maximum 22 km 2000-2007 Mernild et al. 2011 SWE depth Mean ELA Daily ranked runoff (the explained variance) 1% 50 m in elevation 0.86-0.97 2009-2013 Mernild et al. 2015 GrIS SMB Mean annual difference between simulations and observations 0.17 ± 0.23 m w.e. (Forcing and DEM are similar to this TCD study in review) 1979-2014 Mernild et al. 2018

References: Mernild, S. H., Liston, G. E., Hiemstra C. A., and Steffen, K. 2008. Surface Melt Area and Water Balance Modeling on the Greenland Ice Sheet 1995–2005. Journal of Hydrometeorology, 9(6), 1191–1211.

Mernild, S. H., Liston, G. E., van As, D., Hasholt, B., and Yde, J. C. 2018. High-resolution ice sheet surface mass-balance and spatiotemporal runoff simulations: Kangerlussuaq, West Greenland. Arctic, Antarctic, and Alpine Research (Special Issue), doi.org/10.1080/15230430.2017.1415856.

Mernild, S. H., Mote, T., and Liston, G. E. 2011. Greenland Ice Sheet surface melt extent and trends, 1960–2010. Journal of Glaciology, 57(204), 621–628.

Mernild, S. H., Holland, D. M., Holland, D., Rosing-Asvid, A., Yde, J. C., Liston, G. E., and Steffen, K. 2015. Freshwater flux and spatiotemporal simulated runoff variability into Ilulissat Icefjord, West Greenland, linked to salinity and temperature observations near tidewater glacier margins obtained using instrumented ringed seals. Journal of Physical Oceanography, 45(5), 1426–1445, doi: 10.1175/JPO-D-14-0217.1.

Fig 2: showing mean SMB without a validation or without a 2D comparison with other models (e.g. MAR or RACMO) is not interesting for me. Authors: We prefer comparing SnowModel simulations against observations rather than comparing the simulations against other model systems (see above). Comparisons between models may be flawed if both/all models are 'wrong' or biased but still produce similar results. Therefore, it is more comprehensive to use observations for comparison. However, to approach the reviewer's request for model comparisons, we used model simulations from other studies for evaluations in the Result and discussion.

Fig 3b: what is the interest of showing the 2 last figures? Authors: It is simply to highlight the spatial variability in refreezing and retention when using the standard 6-catchment division compared to our 3000+ catchment division. Studies use this simple division of six GrIS basins, but to give a more detailed and spatial catchment illustration of the refreezing and retention conditions we included these last two figures. This was one of the issues, among many, discussed at a Greenland Ice Sheet Retain Workshop, held in June 2016 at GEUS in Copenhagen, and which would be of interest for the community.

Fig5: these figures are not readable. Authors: Please see our response to Reviewer #1 regarding Figure 5.

[Figure]

---

## Author Comment (AC3) · 10 Apr 2018

[revised manuscript text omitted]
 on a 5-km grid using MicroMet. The 6-hour data were scaled to 3-hours by linear interpolation, and the 12-hour precipitation was equally distributed over the 3-hour intervals for the last 12 hours. The 3-hour temporal resolution was chosen to allow SnowModel to resolve the solar radiation diurnal cycle in its simulation of snow and ice temperature evolution and melt processes.

194 The DEM was obtained from Levinsen et al. (2015) (original resolution  $2 \times 2$  km; 4 195  $km^2$ ), and rescaled to a 5-km horizontal grid resolution that covered the GrIS (1,646,175  $km^2$ ), 196 mountain glaciers, and the entire Greenland  $(2,166,725 \text{ km}^2)$  and the surrounding fjords and seas 197 (Figure 1a). The DEM is time-invariant specific to the year 2010. The DEM was developed by 198 merging contemporary radar and laser altimetry data, where radar data were acquired with 199 Envisat and CryoSat-2, and laser data with the Ice, Cloud, and land Elevation Satellite (ICESat), 200 the Airborne Topographic Mapper (ATM), and the Land, Vegetation, and Ice Sensor (LVIS). 201 Radar data were corrected for horizontal, slope-induced, and vertical errors from penetration of 202 the echoes into the subsurface (Levinsen et al. 2015). Since laser data are not subject to such 203 errors, merging radar and laser data yields a DEM that resolves both surface depressions and 204 topographic features at higher altitudes (Levinsen et al. 2015). The distribution of glacier cover 205 was obtained from the Randolph Glacier Inventory (RGI, v. 5.0) polygons; these data were 206 resampled to the 5-km grid. The SnowModel land-cover mask defined glaciers to be present 207 when individual grid cells were covered by 50 % or more with glacier ice.

In MicroMet, only one-way atmospheric coupling was provided, where the meteorological conditions were prescribed at each time step. In the natural system, the atmospheric conditions would be adjusted in response to changes in surface conditions and

211 properties (Liston and Hiemstra 2011). Due to the use of the 5-km horizontal grid increments, 212 snow transport and blowing-snow sublimation processes (usually produced by SnowTran-3D in 213 SnowModel) were excluded from the simulations because blowing snow does not typically move 214 completely across 5-km distances (Liston and Sturm 2002; Mernild et al. 2017). Static 215 sublimation was, however, included in the model integrations. In HydroFlow, the generated 216 catchment divides and flow network were controlled by the DEM, i.e., exclusively by the surface 217 topography and not by the development of the glacial drainage system. The role of GrIS bedrock 218 topography on controlling the potentiometric surface and the associated meltwater flow direction 219 was assumed to be a secondary control on discharge processes (Cuffey and Paterson 2010). 220 First, the GrIS DEM was initially divided into six major sections following Rignot et al.

(unpublished): southwest (SW), west (W), northwest (NW), north (N), northeast (NE), and
southwest (SW) (Figure 1b and Table 1). Second, HydroFlow divided Greenland into 3,272
individual catchments (Figure 1c), each with an eight-compass-direction water-flow network
where water is transported through this network via linear reservoirs. Only a single outlet into the
seas was allowed for each individual catchment.

The mean and median catchment sizes were  $680 \text{ km}^2$  and  $75 \text{ km}^2$ , respectively. The top 226 227 one percent of the largest catchments accounted for 53 % of the Greenland area. This distribution 228 of HydroFlow-defined GrIS catchments (Figure 1c) closely matched both the catchment 229 distribution by Mernild and Liston (2012) and by Rignot and Kanagaratnam (2006) for the 20 230 largest GrIS catchments (not including midsize and minor catchments), both with respect to size 231 and location of the watershed divide. The total number of HydroFlow-generated catchments 232 presented in this study was ~4 % higher due to the use of the DEM obtained from Levinsen et al. 233 (2015), than the number of Greenland catchments in the Mernild and Liston (2012) study.

234 An example of the HydroFlow generated catchment divides and flow network is 235 illustrated in detail by Mernild et al. (2018; Figure 1c) for the Kangerlussuaq catchment in 236 central west Greenland (67°N, 50°W; SW sector of the GrIS): The same catchment from where 237 SnowModel/HydroFlow was evaluated against independent observations (see Section 2.3). 238 Because the DEM is time-invariant, no changes from a thinning ice, ice retreat, and from 239 changes in hypsometry will influence the catchment divides and the flow network patterns, 240 including the glacial drainage system. Changes in runoff over time are therefore solely 241 influenced by the climate signal and the surface snow and ice cover conditions (runoff was 242 generated from gridded inputs from rain, snowmelt, and ice melt). In HydroFlow, the meltwater 243 flow velocities were obtained from dye tracer experiments conducted both through the snowpack 244 (in early and late-summer) and through the englacial and subglacial environments (Mernild et al. 245 2006b).

246

**247 2.3 Evaluation**

248 For Greenland, long-term catchment river runoff observations are sparse; at present at 249 least eight permanent hydrometric monitoring stations are operating (Mernild 2016), measuring 250 the sub-daily and sub-seasonal runoff variability originating from rain, melting snow, and 251 melting ice from local glaciers and the GrIS. In addition, these observations only span parts of 252 the runoff season, ranging between few weeks to approximately three months. For the 253 Kangerlussuaq area, independent meteorological, snow and ice observational, and river runoff 254 datasets are also available, e.g., K-transect point observed air temperature, and SMB and 255 catchment outlet observed runoff (discharge) from Watson River (e.g., van de Wal et al. 2005; 256 van den Broeke et al. 2008a; 2008b, Hasholt et. al. 2013, van As et al. 2018). These observed

[revised manuscript text omitted]

The GrIS SMB for the 35-year mean was  $253.4 \pm 121.4$  Gt yr-1, indicating a negative sea-467 level contribution, and  $135.5 \pm 98.2$  Gt yr-1 for 2005–2014, indicating a trend towards a less 468 469 positive SMB value (Table 1). This change in SMB between the two periods was mainly due to 470 an increase in runoff of 106.7 Gt yr-1, where other water balance components showed relatively 471 lesser increases. For comparison e.g., Vizcaino et al. (2013), Noël et al. (2016), and Wilton et al. (2016) estimated the mean GrIS SMB to be  $359.3 \pm 120$  Gt yr-1 (1960–2005), 349.3 Gt yr-1 472 (1958-2015), and  $382 \pm 78$  Gt yr-1 (1979-2012), respectively. For the GrIS, the 35-year mean 473 474 SMB was negative for the northern region, in balance for northeast Greenland, and positive for 475 all other regions and only positive for the southeastern, southwestern, and western sectors for 476 2005–2014 (Table 1).

[revised manuscript text omitted]
| 841 | on Mittivakkat Gletscher, Southeast Greenland, and the Greenland Ice Sheet. Doctoral thesis.       |
| 842 | Faculty of Science, University of Copenhagen, Copenhagen, pp. 419.                                 |
| 843 |                                                                                                    |
| 844 | Mernild, S. H., B. Hasholt and G. E. Liston 2006b. Water flow through Mittivakkat Glacier,         |
| 845 | Ammassalik Island, SE Greenland. Geografisk Tidsskrift-Danish Journal of Geography, 106(1),        |
| 846 | 25–43.                                                                                             |
| 847 |                                                                                                    |
| 848 | Mernild, S. H., Hanna, E., McConnell, J. R., Sigl, M., Beckerman, A. P., Yde, J. C., Cappelen, J., |
| 849 | and Steffen, K. 2015. Greenland precipitation trends in a long-term instrumental climate context   |
| 850 | (1890–2012): Evaluation of coastal and ice core records. International Journal of Climatology,     |
| 851 | 35, 303–320, doi:10.1002/joc.3986.                                                                 |
| 852 |                                                                                                    |

[revised manuscript text omitted]

- Sugden, D.E., Clapperton, C. M, and Knight, P. G.1985. A jökulhlaup near Søndre Strømfjord,
  West Greenland, and some effects on the ice-sheet margin. Journal of Glaciology, 31(109), 366–
  368.
  Steger, C. R., Reijmer, C. H., van den Broeke, M. R., Wever, N., Forster, R. R., Koenig, L. S.,
  Kuipers Munneke, P., Lehning, M., Lhermitte, S., Ligtenberg, S. R. M., Miège, C. and Noël, B.
  P. Y. 2017. Firn Meltwater Retention on the Greenland Ice Sheet: A Model Comparison. Front.
  Earth Sci. 5:3. doi: 10.3389/feart.2017.00003.
- 954
- Steffen, K. 1995. Surface energy exchange at the equilibrium line on the Greenland ice sheet
  during onset of melt. Annals of Glaciology, 21, 13–18.
- 957
- 958 Straneo, F., Curry, R. G., Sutherland, D. A., Hamilton, G. S., Cenedese, C., Våge, K., and Sterns,
- L. A. 2011. Impact of fjord dynamics and glacial runoff on the circulation near Helheim Glacier.
  Nat.Geosci., 4, 322–327, doi:10.1038/ngeo1109.
- 961
- 962 Tedesco, M., Willis, I. C., Hoffman, M. J., Banwell, A. F., Alexander, P., and Arnold, N. S.
- 963 2013. Ice dynamic response to two modes of surface lake drainage on the Greenland ice sheet.
- 964 Environ. Res. Lett., 8(3), 34007, doi:10.1088/1748-9326/8/3/034007.
- 965
- 966 Tedesco, M., Box, J. E., Cappelen, J., Fettweis, X., Mote. T., van de Wal, R. S. W., Smeets, C. J.
- P. P., and Wahr, J. 2014. Greenland Ice Sheet. In Jeffries, M. O., Richter-Menge, J. A., and
- 968 Overland, J. E. (eds.). Arctic Report Card 2014.
- 969
- 970 Tedesco, M., Box, J. E., Cappelen, J., Fausto, R. S., Fettweis, X., Mote, T., Smeets C. J. P. P.,
- 971 van As, D., Velicogna, I., van de Wal, R. S. W. and Wahr, J. 2016. Greenland Ice Sheet. In
- 972 Richter-Menge, J. A., Overland, J. E., and Mathis, J. T. (eds.). Arctic Card Report 2016.
- 973
- 974 Uppala, S. M., Kållberg, P. W., Simmons, A. J., Andrae, U., Da Costa Bechtold, V., Fiorino, M.,
- 975 Gibson, J.K., Haseler, J., Hernandez, A., Kelly, G. A., Li, X., Onogi, K., Saarinen, S., Sokka,

- 976 N., Allan, R. P., Anderson, E., Arpe, K., Balmaseda, M. A., Beljaars, A. C. M., Van De Berg, L.,
- 977 Bidlot, J., Bormann, N., Caires, S., Chevallier, F., Dethof, A., Dragosavac, M., Fisher, M.,
- 978 Fuentes, M., Hagemann, S., Hólm, E., Hoskins, B. J., Isaksen, L., Janssen, P. A. E. M., Jenne, R.,
- 979 Mcnally, A. P., Mahfouf, J.-F., Morcrette, J.-J., Rayner, N. A., Saunders, R.W., Simon, P., Sterl,
- A., Trenbreth, K. E., Untch, A., Vasiljevic, D., Viterbo, P., and Woollen, J.: The ERA-40 re-
- 981 analysis, Q. J. Roy. Meteor. Soc., 131, 2961–3012, doi:10.1256/qj.04.176, 2005.
- 982
- van Angelen, J. H., Lenaerts, J. T. M., van den Broeke, J. T. M., Fettweis, X., van Meijgaard, E.
- 2013. Rapid loss of firn pore space accelerates 21 century Greenland mass loss. Geophysical
- 985 Research Letter, 40, 2109–2113.
- 986
- van As., D, Box, J. E., and Fausto, R. S., 2016: Challenges of Quantifying Meltwater Retention
- 988 in Snow and Firn: An Expert Elicitation. Frontiers in Earth Science, 4(101), 1–5.
- 989
- van As, D. Hasholt, B., Ahlstrøm, A. P., Box, J. E., Cappelen, J., Colgan, W., Fausto, R. S.,
- 991 Mernild, S. H., Mikkelsen, A.B., Noël, B. P.Y., Petersen, D., and Van den Broeke, M. R. 2018.
- 992 The longest observationally-constrained record of Greenland ice sheet meltwater discharge
- 993 (1949–2016). Accepted, Arctic, Antarctic, and Alpine Research (Special Issue).
- van den Broeke, M. R., Enderlin, E. M., Howat, I. M., Munneke, P.K., Noël, B. P., Y., van de
- Berg, W. J., van Meijgaard, E., and Wouters, B. 2016. On the recent contribution of the
- 996 Greenland ice sheet to sea level change. The Cryosphere, 10, 1933–1946, doi:10.5194/tc-10-
- 997 1933-2016.

- 999 van den Broeke, M., Smeets, P., Ettema, J., and Munneke, P. K. 2008a. Surface radiation balance
- 1000 in the ablation zone of the west Greenland ice sheet, J. Geophys. Res., 113, D13105,
- 1001 doi:10.1029/2007/JD009283.
- 1002
- 1003 van den Broeke, M., Smeets, P., Ettema, J., van der Veen, C., van de Wal, R., and Oerlemans, J.
- 1004 2008b. Partitioning of melt energy and meltwater fluxes in the ablation zone of the west
- 1005 Greenland ice sheet, The Cryosphere, 2, 179–189, doi:10.5194/tc-2-179-2008.
- 1006

- 1007 van de Wal, R. S. W., Greuell, W., van den Broeke, M. R., Reijmer, C. H., and Oerlemans, J.
- 1008 2005. Surface mass-balance observations and automatic weather station data along a transect
- 1009 near Kangerlussuaq, West Greenland, Ann. Glaciol., 42, 311–316.
- 1010
- 1011 van de Wal, R. S. W., Boot, W., van den Broeke, M., Smeets, C. J. P. P., Reijmer, C. H., Donker,
- 1012 J. J. A., and Oerlemans, J. 2008. Large and rapid melt-induced velocity changes in the ablation
- 1013 zone of the Greenland ice sheet. Science, 321, 111–113.
- 1014
- 1015 van der Veen, C. J. 2007. Fracture propagation as a means of rapidly transferring surface
- 1016 meltwater to the base of glaciers. Geophys. Res. Lett., 34, L01501.
- 1017
- 1018 Weijer, W., M. E. Maltrud, M. W. Hecht, H. A. Dijkstra, and M. A. Kliphuis, 2012. Response of
- 1019 the Atlantic Ocean circulation to Greenland Ice Sheet melting in a strongly-eddying ocean

1020 model. Geophys. Res. Lett., 39, L09606, doi:10.1029/2012GL051611.

- 1021
- 1022 Wilton, D. J., Jowett, A., Hanna, E., Bigg, G. R., van den Broeke, M. R., Fettweis, X., and
- 1023 Huybrechts, P., 2017. High resolution (1 km) positive degree-day modelling of Greenland ice
- sheet surface mass balance, 1870-2012 using reanalysis data. Journal of Glaciology, 63(237),
  1025 176–193.
- 1026
- 1027 Vizcaino, M., Lipscomb, W. H., Sacks, W. J., van Angelen, J. H., Wouters, B., and van den
- 1028 Broeke, M. R. 2013. Greenland Surface Mass Balance as Simulated by the Community Earth
- 1029 System Model. Part I: Model Evaluation and 1850-2005 Results. Journal of Climate, 26, 7793–
- 1030 7812, doi.10.1175/JCLI-D-12-00615.1.
- 1031
- 1032 Yang, D., Ishida, S., Goodison, B. E., and Gunter, T. 1999. Bias correction of daily precipitation
- 1033 measurements for Greenland. J. Geophys. Res. 104(D6), 6171–6181,
- 1034 doi:10.1029/1998JD200110.
- 1035
- 1036 Zwally, H. J., Abdalati, W., Herring, T., Larson, K., Saba, J., and Steffen, K. 2002. Surface melt-
- 1037 induced acceleration of Greenland ice-sheet flow. Science, 297, 218–222.
  - 40